# Palmitoylation of LIM Kinase-1 ensures spine-specific actin polymerization and morphological plasticity

Joju George[1,2], Cary Soares[3,4,5], Audrey Montersino[1,2], Jean-Claude Beique[3,4,5], Gareth M Thomas[1,2]*

[1]Shriners Hospitals Pediatric Research Center, Temple University School of Medicine, Philadelphia, United States; [2]Department of Anatomy and Cell Biology, Temple University School of Medicine, Philadelphia, United States; [3]Heart and Stroke Partnership for Stroke Recovery, University of Ottawa, Ottawa, Canada; [4]Center for Neural Dynamics, Department of Cellular and Molecular Medicine, University of Ottawa, Ottawa, Canada; [5]Department of Cellular and Molecular Medicine, University of Ottawa, Ottawa, Canada

**Abstract** Precise regulation of the dendritic spine actin cytoskeleton is critical for neuro-development and neuronal plasticity, but how neurons spatially control actin dynamics is not well defined. Here, we identify direct palmitoylation of the actin regulator LIM kinase-1 (LIMK1) as a novel mechanism to control spine-specific actin dynamics. A conserved palmitoyl-motif is necessary and sufficient to target LIMK1 to spines and to anchor LIMK1 in spines. ShRNA knockdown/rescue experiments reveal that LIMK1 palmitoylation is essential for normal spine actin polymerization, for spine-specific structural plasticity and for long-term spine stability. Palmitoylation is critical for LIMK1 function because this modification not only controls LIMK1 targeting, but is also essential for LIMK1 activation by its membrane-localized upstream activator PAK. These novel roles for palmitoylation in the spatial control of actin dynamics and kinase signaling provide new insights into structural plasticity mechanisms and strengthen links between dendritic spine impairments and neuropathological conditions.

*For correspondence: gareth.thomas@temple.edu

## Introduction

Most excitatory synapses are formed on dendritic spines—small protrusions that decorate the shaft of neuronal dendrites (*Bourne and Harris, 2008*; *Hotulainen and Hoogenraad, 2010*). Changes in the size and shape of individual spines are closely associated with Long-term potentiation (LTP), a cellular correlate of learning and memory (*Fifková and Van Harreveld, 1977*; *Yuste and Bonhoeffer, 2001*; *Matsuzaki et al., 2004*; *Bosch and Hayashi, 2012*; *Murakoshi and Yasuda, 2012*). Moreover, abnormal spine morphology and/or density are hallmarks of Intellectual Disability and other cognitive dysfunctions, including Autism-Spectrum Disorders and schizophrenia (*Fiala et al., 2002*; *Penzes et al., 2011*). These findings suggest that precise regulation of dendritic spine morphology and number is critical for normal cognition.

Spines are highly enriched in actin filaments, and dynamic modulation of the actin cytoskeleton is crucial for controlling not only spine formation and elimination, but also modifications of the size, shape and motility of existing spines (*Hotulainen and Hoogenraad, 2010*; *Bosch and Hayashi, 2012*). In response to local synaptic cues, neurons can rapidly alter the morphology of individual spines (*Matsuzaki et al., 2004*; *Okamoto et al., 2004*; *Honkura et al., 2008*; *Murakoshi et al., 2011*). This process likely requires precise spatial regulation of proteins that increase actin polymerization and those that sever/disassemble actin

**eLife digest** Neurons transmit information from one cell to the next by passing signals across junctions called synapses. For the neurons that receive these signals, these junctions are found on fine branch-like structures called dendrites that stick out of the cell. Dendrites themselves are decorated with smaller structures called dendritic spines, which typically receive information from one other neuron via a single synapse. Dendritic spines form in response to the signaling activity of the neuron, and problems with forming these spines have been linked to conditions such as autism and schizophrenia.

Dendritic spines are created by the cell's cytoskeleton—a network of proteins that creates a constantly changing internal scaffold that shapes cells. One cytoskeleton protein called actin exists as thin filaments that can be extended or broken up by other proteins. It is not fully understood how actin is regulated in the dendritic spines. However, some researchers thought that the proteins that control the formation of the actin filaments would need to be localized to the dendritic spines to ensure that the spines form correctly.

Some proteins can be made to localize to cell membranes by attaching a molecule called palmitic acid to them. Previous research has suggested that this 'palmitoylation' process is particularly important in neurons. Through a combination of experimental techniques, George et al. now show that palmitoylation is required to localize a protein called LIMK1, which regulates the construction of actin filaments, to the tips of dendritic spines. Further experiments showed that blocking the palmitoylation of LIMK1 alters how actin filaments form, makes spines unstable and causes synapses to be lost.

George et al. also discovered that palmitoylation is necessary for LIMK1 to be activated by another protein that is found at dendritic spine membranes. This 'dual-control' mechanism makes it possible to precisely control where actin filaments form within dendritic spines. In addition to LIMK1, several other enzymes are also modified by palmitoylation. It will therefore be interesting to determine whether this dual control mechanism is broadly used by neurons to precisely regulate the structure and function of individual spines and synapses.

filaments in a given spine. However, many actin regulators are predicted to be soluble, diffusible proteins, which appear poorly suited to operate with the necessary spatial specificity. This raises the question of how spatially precise, spine-specific actin regulation is achieved.

We hypothesized that neurons must possess mechanisms to localize and/or confine certain actin regulatory proteins within dendritic spines, not only to ensure spine-specific regulation but perhaps also to control actin polymerization/depolymerization at the subspine level. One mechanism to control protein localization is palmitoylation, a protein-lipid modification that targets cytosolic proteins to specific membranes (*Fukata and Fukata, 2010*; *Thomas and Huganir, 2013*). Palmitoylation occurs in all eukaryotic cells but appears to be especially important in neurons because human genetic mutations and mouse knockouts of Palmitoyl acyltransferases (PATs, which catalyze palmitoylation) frequently lead to neurological and/or cognitive deficits (*Fukata and Fukata, 2010*; *Greaves and Chamberlain, 2011*). We therefore hypothesized that palmitoylation might modulate actin regulators to ensure spatially restricted signaling in spines.

Two important actin regulatory proteins are the LIM kinases (LIMK1 and LIMK2 [*Tada and Sheng, 2006*]), which phosphorylate and inactivate the actin severing protein cofilin, thus promoting actin polymerization (*Mizuno et al., 1994*; *Arber et al., 1998*; *Yang et al., 1998*). Interestingly, LIMK1 appears particularly important for actin regulation in spines, because *LIMK1* knockout in mice or genetic mutation in humans is associated with spine abnormalities and cognitive impairments (*Frangiskakis et al., 1996*; *Tassabehji et al., 1996*; *Meng et al., 2002*). Moreover, although LIMK1's upstream activator in neurons is unclear, spine abnormalities and intellectual impairments are also linked to mutations in *PAK3* (*Allen et al., 1998*; *Boda et al., 2004*), a member of the p21-activated kinase (PAK) family that phosphorylates and activates LIMK1 in non-neuronal cells (*Edwards et al., 1999*). These findings link PAK/LIMK1-dependent regulation of actin polymerization to the control of spine morphology and higher brain function. However, PAKs and LIMKs are predicted soluble, cytosolic proteins. How, then, might they regulate actin with the necessary spatial precision?

Here we report that palmitoylation of LIMK1 at a specific N-terminal motif is necessary and sufficient to target LIMK1 to spines. Palmitoyl-LIMK1 is essential for normal spine actin turnover, activity-dependent morphological plasticity and long-term spine stability. Strikingly, palmitoylation controls not only LIMK1 localization but also its activation by PAK in neurons. This novel 'dual-control' mechanism ensures spatially precise actin regulation at the single spine and potentially also the subspine levels.

## Results

### A conserved palmitoyl-motif in LIMK1

To address whether palmitoylation of actin regulators facilitates spatial control of actin polymerization in spines, we performed two bioinformatic searches for potential palmitoyl-motifs in actin regulatory proteins. LIMK1 was a prominent hit in both searches and adjacent cysteine residues (Cys7, Cys8, conserved in all vertebrate LIMK1 orthologs) were predicted to be palmitoylated (*Figure 1A*).

To test whether the LIMK1 Cys7/Cys8 (CC) motif is indeed palmitoylated, we transfected HEK293T cells with C-terminal myc-tagged wild type LIMK1 (wt-LIMK1-myc) or CCSS-LIMK1-myc (Cys7/Cys8 mutated to non-palmitoylatable Ser). To isolate palmitoyl-proteins, we subjected lysates to Acyl-biotin exchange (ABE). ABE uses an exchange of thioester-linked acyl modifications (i.e., palmitoylation), for biotin, with the resultant biotinylated proteins being affinity-purified using neutravidin-conjugated beads (*Wan et al., 2007*; *Thomas et al., 2012*). Wt-LIMK1-myc was clearly detected in ABE fractions, indicative of palmitoylation, but was not detected in control purifications in which the essential ABE reagent hydroxylamine ($NH_2OH$) was omitted (*Figure 1B*). In contrast, CCSS-LIMK1-myc was not detected in ABE fractions (*Figure 1B*), suggesting that LIMK1 is palmitoylated at Cys7/Cys8.

### Palmitoylation is necessary to target and confine LIMK1 to dendritic spines

To investigate the neuronal role of palmitoyl-LIMK1, we examined LIMK1 palmitoylation in cultured hippocampal neurons, focusing on four developmental stages: 4 days in vitro (DIV4), when neurites are extending; DIV8, when dendrites elaborate and undergo branching; DIV12, when dendritic spines first appear and DIV20, when spines and synapses are mature (a time course similar to previous reports [*Kaech and Banker, 2006*; *Beaudoin et al., 2012*]). Interestingly, despite similar LIMK1 protein expression across these developmental stages, palmitoyl-LIMK1 was not detected in ABE fractions from DIV4 neurons (*Figure 1C*). However, palmitoyl-LIMK1 became detectable at DIV8, increased markedly at DIV12 and remained prominent at DIV20 (*Figure 1C*). These findings suggest that LIMK1 is palmitoylated in hippocampal neurons and that its palmitoylation coincides with spine development and maturation. This conclusion is consistent with the robust palmitoylation of LIMK1 in adult rat forebrain homogenates (*Figure 1D*).

The temporal correlation of LIMK1 palmitoylation and spine formation (*Figure 1C*), and the altered spine morphology in *LIMK1* knockout mice (*Meng et al., 2002*), suggested that palmitoyl-LIMK1 might play a specific role in spines. In support of this hypothesis, wt-LIMK1 was highly enriched in spines but CCSS-LIMK1 was not (*Figure 1E,F*). We reasoned that palmitoylation might anchor LIMK1 in spines to limit its diffusion, thus enhancing spatial control of actin regulation. We therefore transfected neurons with GFP-tagged LIMK1 and monitored its diffusibility by Fluorescence Recovery After Photobleaching (FRAP) using time-lapse imaging. Prominent FRAP, indicative of diffusion of the tagged protein, was observed with CCSS-LIMK1-GFP but not with wt-LIMK1-GFP (*Figure 1—figure supplement 1*). Quantitative analysis of FRAP data confirmed that CCSS mutation decreased the stable fraction of LIMK1-GFP in spines (*Figure 1—figure supplement 1*). We note that the time course of LIMK1-GFP FRAP, irrespective of its palmitoylation status, is markedly slower than previously characterized soluble proteins (*Star et al., 2002*; *Bingol et al., 2010*; *Zheng et al., 2010*). This suggests that other factors, most likely protein–protein interactions, are important to constrain LIMK1 in spines. However, our FRAP results suggest that palmitoylation, in addition to targeting LIMK1 to spines, also increases the fraction of LIMK1 that is stably anchored in spine heads.

As an additional line of evidence that the impaired spine targeting caused by Cys7,8 mutation is palmitoylation-specific, we used a pharmacological approach. The palmitoylation inhibitor 2-Bromopalmitate (2-Br; [*Jennings et al., 2009*]) did not affect endogenous LIMK1 protein levels,

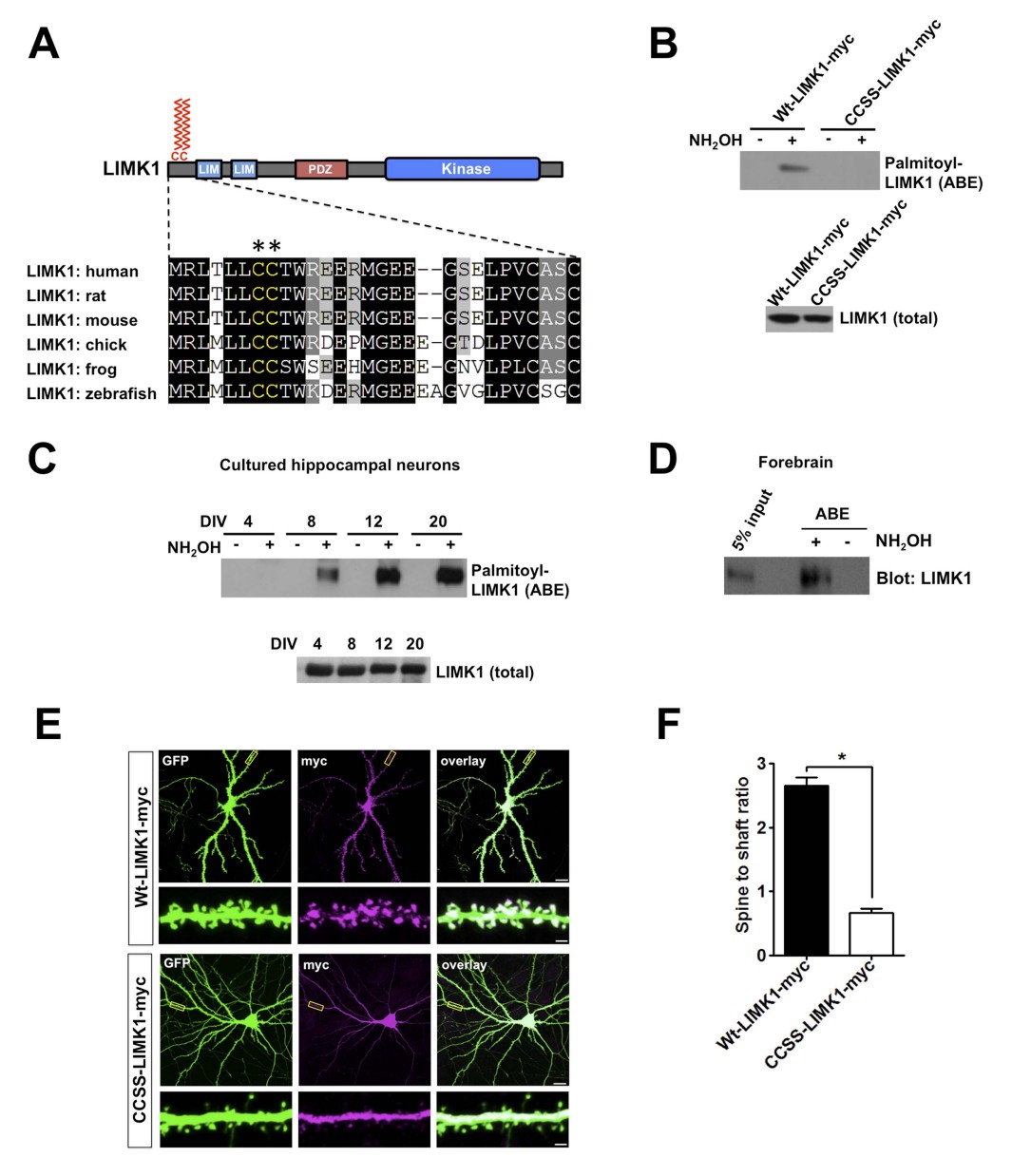

**Figure 1**. Palmitoylation at a unique di-cysteine motif targets LIMK1 to dendritic spines. (**A**). *Upper panel*: LIMK1 schematic, showing predicted palmitoyl-motif (CC, red) and LIM, PDZ and kinase domains. *Lower panel*: Multiple sequence alignment of the N-terminal region of LIMK1 orthologs from the indicated species. CC palmitoylation motif (highlighted with asterisks) is conserved in vertebrates. (**B**) HEK293T cells were transfected with C-terminal myc-tagged LIMK1wt (wt-LIMK1-myc) or CCSS-LIMK1-myc (cys 7, 8 mutated to Ser). ABE fractions prepared from lysates were blotted to detect palmitoyl-LIMK1 (top panel). Lysates were blotted to detect total LIMK1 expression (bottom panel). CCSS mutation eliminates LIMK1 palmitoylation. (**C**) LIMK1 palmitoylation increases, coincident with spine maturation and synapse formation. Hippocampal neurons, cultured for the indicated number of Days in vitro (DIV), were lysed and identical amounts of total protein were subjected to ABE to detect palmitoyl-LIMK1. (**D**) Homogenate and palmitoylated (ABE) fractions from rat forebrain were blotted to detect LIMK1.
(**E**) Palmitoylation targets LIMK1 to dendritic spines. Representative images of hippocampal neurons (DIV18), transfected to express GFP plus Wt-LIMK1-myc or CCSS-LIMK1-myc and immunostained with the indicated antibodies. Scale bar: 20 μm. Lower panels show magnified images of single dendrites (scale bar: 1 μm).
(**F**) Quantified spine targeting ratio (signal intensity in dendritic spines compared to adjacent dendritic shaft) for

*Figure 1. continued on next page*

*Figure 1. Continued*

each construct from **E**. Data are mean + SEM for n = 30 neurons for each condition. *p < 0.05, ANOVA with Dunnett's *post hoc* correction.

The following figure supplements are available for figure 1:

**Figure supplement 1**. Palmitoylation helps to anchor LIMK1 in spines.

**Figure supplement 2**. Further evidence that palmitoylation targets LIMK1 to spines.

but greatly reduced LIMK1 palmitoylation in neurons (*Figure 1—figure supplement 2*), reinforcing the conclusion that LIMK1 is palmitoylated. In parallel experiments, 2-Br greatly reduced wt-LIMK1 spine targeting without affecting overall spine numbers (*Figure 1—figure supplement 2*). The similar effects of CCSS mutation and 2-Br treatment strongly suggest that Cys7/Cys8 palmitoylation is required for LIMK1 spine targeting.

## LIMK1's dual palmitoylation motif is sufficient for spine targeting

We next asked whether LIMK1's palmitoylation motif is sufficient for spine targeting. LIMK1-GFP deletion mutants lacking the kinase domain (LIMK1-1-258) or lacking the kinase and PDZ domains (LIMK1 1-137) localized to spines as effectively as full-length LIMK1 (*Figure 2A–C*). Remarkably, a minimal palmitoylation motif construct lacking the kinase, PDZ and LIM domains (LIMK1 [1–15]-GFP; *Figure 2A–C*) was sufficient to target GFP to spines. In contrast, cytosolic GFP was not enriched in spines (*Figure 2A–C*). These findings indicate that LIMK1's N-terminal 15 amino acid sequence is a specific spine targeting motif.

Other post-synaptic palmitoyl-proteins contain di-cysteine motifs (*Kang et al., 2008*; *Fukata and Fukata, 2010*; *Brigidi et al., 2014*), raising the possibility that LIMK1 must be dually palmitoylated to localize to spines. Consistent with this notion, C7S- and C8S-LIMK1 mutants were not targeted to spines (*Figure 2—figure supplement 1*). Moreover, a CCSS-LIMK1 mutant carrying a sequence that directs addition of the lipid myristate (Myr-CCSS-LIMK1), which can mimic a single palmitoylation event (*Thomas et al., 2012*), was not targeted to spines (*Figure 2—figure supplement 1*).

Lastly, we examined the spine targeting of N-terminal mutants of LIMK1 that are predicted to be doubly lipid-modified with myristate plus single palmitate (Myr-SC-LIMK1, Myr-CS-LIMK1). Neither of these mutants was enriched in spines (*Figure 2—figure supplement 2*). Taken together, these results suggest that an intact CC palmitoyl-motif is critical for spine LIMK1 targeting.

## LIMK1's palmitoyl-motif accounts for differential localization of LIMK1 and LIMK2

Interestingly, *LIMK1* genetic loss or mutation affects spine structure and cognition, despite the presence of LIMK2, which is also expressed in hippocampal neurons (*Cajigas et al., 2012*). However, we hypothesized that LIMK2's lacks of a CC palmitoyl-motif (*Figure 3A*) might render LIMK2 unable to localize to spines and thus unable to compensate for loss of LIMK1. Consistent with this notion, LIMK1 was detected in rat brain Post-Synaptic Density (PSD) fractions, including the PSDIII fraction, which is enriched in spine-associated proteins. In contrast, LIMK2 was absent from PSD fractions (*Figure 3B*, *Figure 3—figure supplement 1*) and was not enriched in spines (*Figure 3C,D*). Biochemical and immunocytochemical assays thus suggest that LIMK1 is targeted to spines but LIMK2 is not. Strikingly, however, a chimeric protein consisting of LIMK1's minimal spine targeting sequence fused to LIMK2 was readily detected in spines (*Figure 3C,D*). This result suggests that LIMK1's palmitoyl-motif accounts for differential localization of the two LIMKs, potentially explaining why loss of LIMK1 causes spine-specific phenotypes.

## Loss of palmitoyl-LIMK1 impacts spine actin turnover

The striking effects of palmitoylation on LIMK1 targeting to, and anchoring in, dendritic spines (*Figure 1*, *Figure 1—figure supplement 1*) suggested that palmitoylation is critical for LIMK1-dependent control of actin polymerization (*Arber et al., 1998*; *Yang et al., 1998*) in spines.

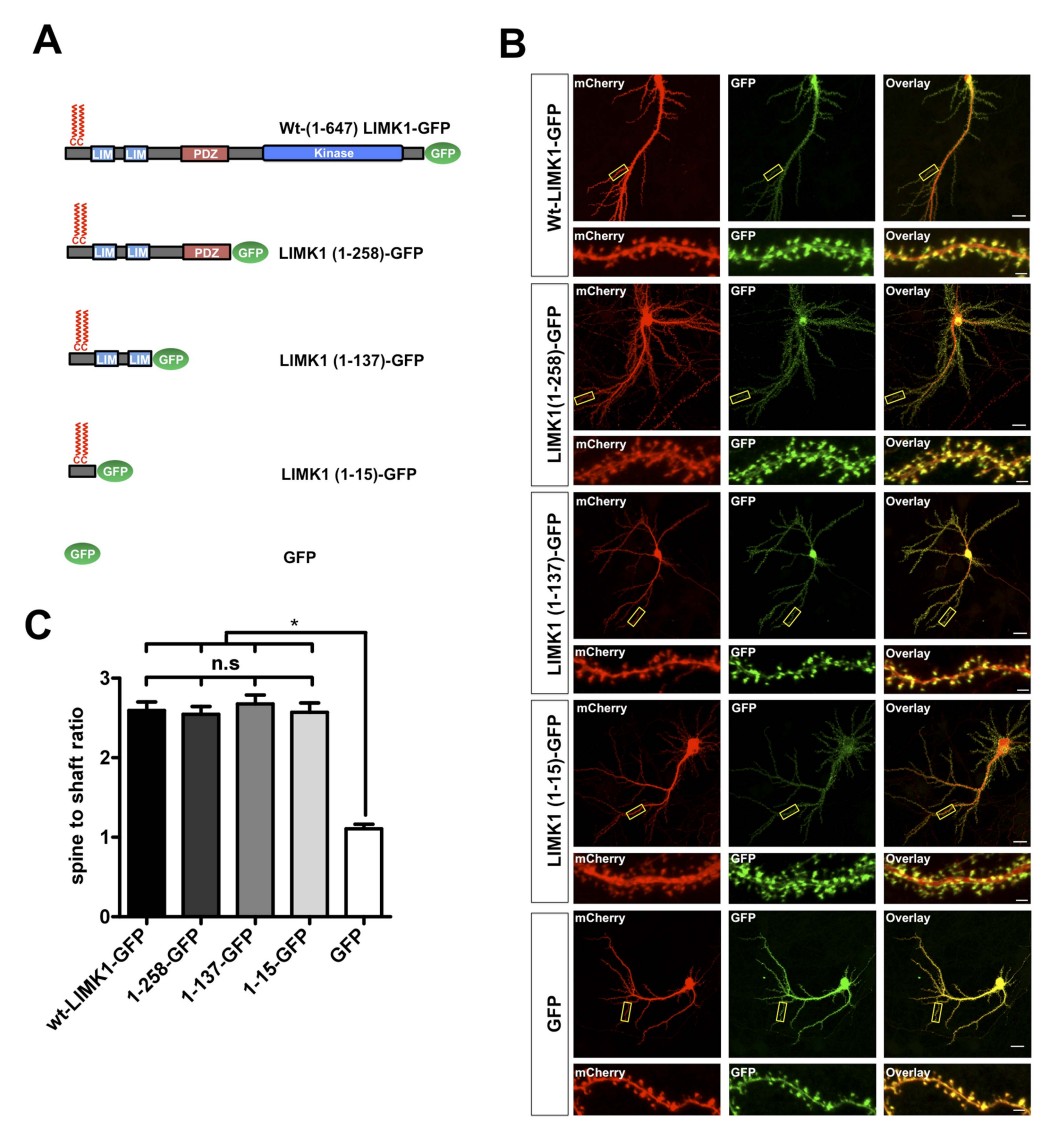

**Figure 2**. LIMK1's palmitoyl-motif is a minimal spine targeting sequence. (**A**) Schematic of LIMK1 deletion mutants. (**B**) LIMK1's palmitoyl-motif is sufficient for spine targeting. Representative images of hippocampal neurons (DIV18), transfected to express mCherry plus the indicated GFP-tagged LIMK1 deletion mutants from **A** and immunostained with the indicated antibodies (scale bar: 20 µm). Lower panels show magnified images of single dendrites (scale bar: 1 µm). (**C**) Spine targeting ratio (mean ± SEM ) for n = 5–15 neurons per condition from **B**. n.s.; p > 0.05 compared to wt-LIMK1-GFP. *; p < 0.05 compared to GFP. ANOVA with Tukey *post hoc* test.

The following figure supplements are available for figure 2:

**Figure supplement 1**. Dual palmitoylation is necessary for LIMK1 spine targeting.

**Figure supplement 2**. LIMK1 mutants that are predicted to be dually lipid modified are not enriched in spines.

To examine this possibility, while circumventing possible issues arising from LIMK1's neuro-developmental roles (*Meng et al., 2002*; *Rosso et al., 2004*), we used small hairpin RNA (shRNA) knockdown/rescue to replace endogenous LIMK1 with the CCSS-LIMK1 mutant in mature neurons. We first identified an shRNA that potently reduced levels of cotransfected rat LIMK1-myc (*Figure 4—figure supplement 1*). The same shRNA greatly reduced endogenous LIMK1 levels when

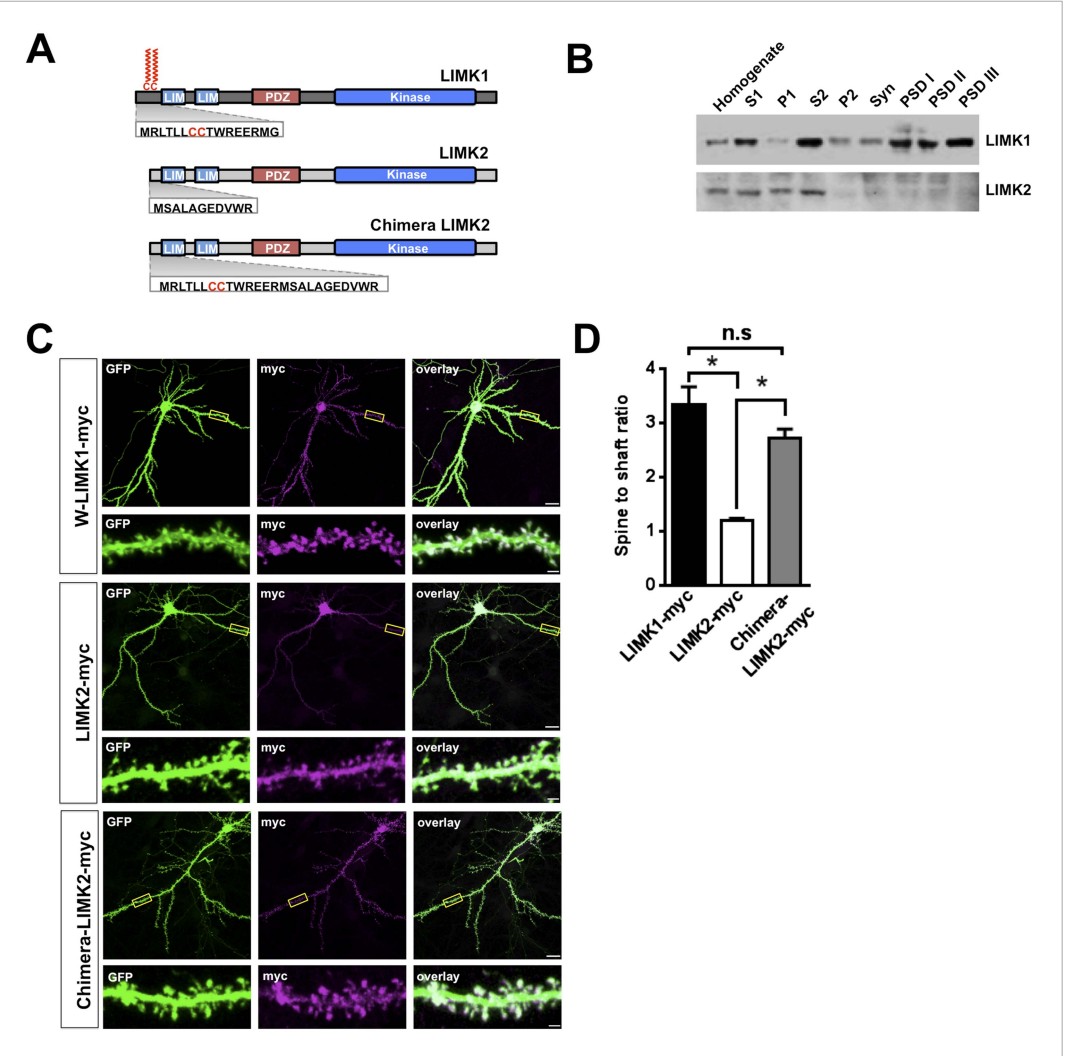

**Figure 3**. LIMK1's palmitoyl-motif accounts for differential localization of LIMK1 and LIMK2. (**A**) Schematic of LIMK1 (top) and LIMK2 (middle), showing similar overall domain arrangement. Rat LIMK1 and LIMK2 kinase domains are 70% identical, 84% similar (NCBI BLAST). Expanded N-terminal sequences show LIMK1's unique CC motif. Lower schematic shows a chimeric protein in which the N-terminal 15 amino acids of LIMK1 are fused to LIMK2 ('Chimera LIMK2'). (**B**) The indicated rat forebrain subcellular fractions were blotted to detect endogenous LIMK1 and LIMK2. LIMK1 is seen in Post-synaptic Density (PSD) fractions, consistent with a previous study, but LIMK2 is not. Fidelity of the preparation and antibody specificity is confirmed in *Figure 3—figure supplement 1*. (**C**) Hippocampal neurons (DIV18) were transfected to express GFP plus the indicated myc-tagged LIMK variants. Representative images of single neurons immunostained with the indicated antibodies are shown (scale bar: 20 µm). Lower panels show magnified images of single dendrites. (**D**) Spine targeting ratio (mean ± SEM) for n = 30 neurons per condition from **C**. * p < 0.05; n.s.: p > 0.05, ANOVA with Dunnett's *post hoc correction*.

The following figure supplement is available for figure 3:

**Figure supplement 1**. Specificity of LIMK1 and LIMK2 antibodies and fidelity of subcellular fractionation.

packaged into lentiviral particles and used to infect hippocampal neurons (*Figure 4—figure supplement 1*). shRNA-resistant (shr) 'rescue' forms of wt-LIMK1 (shr-wt-LIMK1-myc) and CCSS-LIMK1 (shr-CCSS-LIMK1-myc) were insensitive to LIMK1 shRNA (*Figure 4—figure supplement 1*).

To address palmitoyl-LIMK1's role in spine actin regulation, we used FRAP to monitor the dynamics of GFP-tagged beta actin (GFP-actin) in single spines (*Star et al., 2002*; *Okamoto et al., 2004*; *Hotulainen et al., 2009*). Actin filaments in spines normally turn over rapidly, due to

treadmilling, while actin monomers exchange bidirectionally between spines and the adjacent dendritic shaft ([*Star et al., 2002*; *Bosch and Hayashi, 2012*]; *Figure 4A*). If GFP-actin in a spine is photobleached, new fluorescent GFP-actin monomers normally diffuse into the spine and are incorporated into the barbed end of bleached filaments, predominantly in the juxtamembrane ('shell') region of the spine head (*Hotulainen et al., 2009*; *Frost et al., 2010*). Concomitantly, bleached actin molecules are severed from filament pointed ends (closer to the spine 'core') and exchange out of the spine, which recovers its fluorescence. However, if turnover is impaired, bleached GFP-actin remains trapped within filaments and FRAP is attenuated. The fraction of actin filaments in the spine that are undergoing rapid turnover can thus be determined from the extent of FRAP (*Star et al., 2002*).

Live imaging of GFP-actin and cotransfected mCherry in DIV17 hippocampal neurons revealed numerous morphologically mature (stubby or mushroom-shaped) spines containing strong GFP-actin signals (*Figure 4B*). Selective photobleaching of single spines rapidly reduced GFP-actin fluorescence, which recovered to an extent similar to previous reports (*Figure 4C–E*; [*Star et al., 2002*]). LIMK1 'knockdown' neurons expressing GFP-actin, mCherry and LIMK1 shRNA had no change in spine number or morphology in the short term (24 hr post transfection; *Figure 4—figure supplement 1*). However, the recovery of GFP-actin fluorescence in LIMK1 knockdown neurons was significantly attenuated, compared to control neurons. This attenuated recovery was evident in plots of the averaged data (*Figure 4D*), or plots of all individual FRAP measurements (*Figure 4—figure supplement 2*) and is consistent with an increase in the percentage of stable GFP-actin. Two analytical methods confirmed that the fraction of stable actin is indeed significantly increased in LIMK1 knockdown spines (*Figure 4E*, *Figure 4—figure supplement 2*), see also [*Koskinen et al., 2014*]). Strikingly, the increase of stable GFP-actin caused by LIMK1 knockdown was restored to control levels by shr-wt-LIMK1, but not by shr-CCSS-LIMK1 (*Figure 4C–E*, *Figure 4—figure supplement 2*).

In contrast to these marked effects on the pool of stable actin, the half-time of GFP-actin fluorescence recovery was not significantly different under any condition examined (*Figure 4—figure supplement 2*). Taken together, these data suggest that the predominant effect of loss of palmitoyl-LIMK1 on GFP-actin turnover is to reduce the pool of mobile actin in spines. This deficit is initially surprising, because LIMK1 is best known as a negative regulator of cofilin, and increased cofilin activity in the absence of palmitoyl-LIMK1 might be expected to *increase* actin turnover. We consider possible molecular explanations for this finding in the 'Discussion', but taken together, these results suggest that palmitoyl-LIMK1 is essential for normal actin turnover in spines.

## Palmitoyl-LIMK1 is critical for activity-dependent spine enlargement

We next sought to identify functional consequences of impaired spine actin turnover, caused by loss of palmitoyl-LIMK1. Spine morphological plasticity critically requires actin polymerization (*Matsuzaki et al., 2004*; *Okamoto et al., 2004*), and could thus potentially also require palmitoyl-LIMK1. To address this possibility, we used shRNA knockdown/rescue in organotypic hippocampal slices and examined spine-specific morphological plasticity following focal activation of glutamate receptors by 2-photon (2P) uncaging of MNI-glutamate (*Matsuzaki et al., 2004*; *Harvey and Svoboda, 2007*).

In biolistically transfected CA1 pyramidal neurons expressing the morphology marker mCherry, focal uncaging of MNI-glutamate on spines induced a rapid, lasting volume increase of the stimulated spine, similar to published results (135 ± 7% of initial volume, 15–25 min post-uncaging; *Figure 5A,B*, *Figure 5—figure supplement 1*). An analogous uncaging protocol (i.e., pairing to 0 mV in whole-cell recordings) likewise led to a persistent increase in the volume of the stimulated spine that was accompanied by a spine-specific increase in the amplitude of uncaging-evoked postsynaptic currents (uEPSCs; *Figure 5—figure supplement 2*).

In subsequent experiments we focused on activity-dependent spine enlargement and first addressed the requirement for LIMK1 in this process. In CA1 pyramidal neurons expressing mCherry plus LIMK1 shRNA, activity-dependent spine enlargement was significantly attenuated (116 ± 6% of initial volume, 15–25 min post-uncaging; p = 0.032 vs control neurons, Mann–Whitney U test; *Figure 5A,B*, *Figure 5—figure supplement 1*). This result suggests that LIMK1 is required for activity-dependent spine enlargement, consistent with the importance of actin polymerization in this process (*Matsuzaki et al., 2004*; *Okamoto et al., 2004*). Strikingly, activity-dependent spine enlargement could be rescued by shr-wt-LIMK1 (control: 136 ± 6%; wt-LIMK1 rescue: 140 ± 8%, p = 0.496,

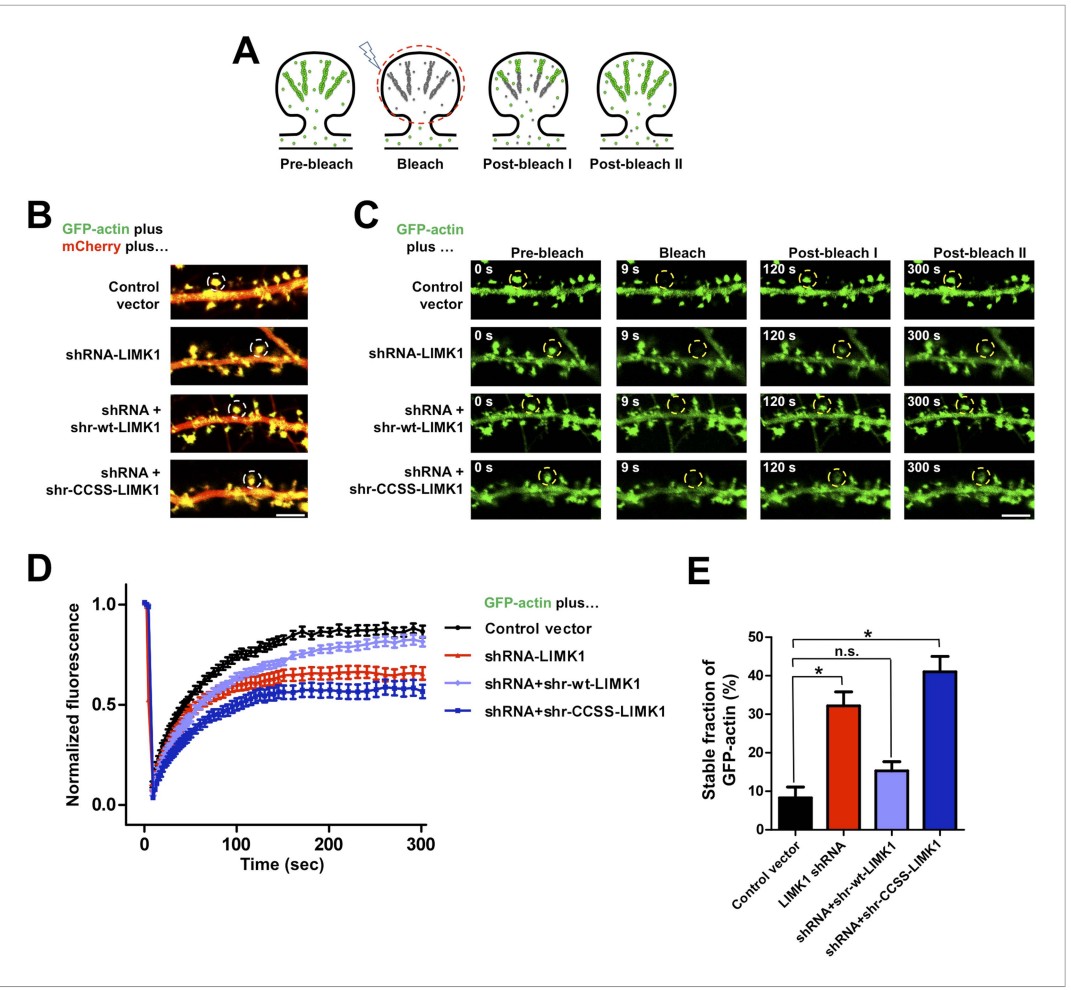

**Figure 4**. Acute loss of palmitoyl-LIMK1 impairs Fluorescence Recovery After Photobleaching (FRAP) of GFP-actin in dendritic spines. (**A**) Schematic of FRAP assay. GFP-actin is photobleached in a single spine. When actin filament turnover is normal, fluorescence recovers as new fluorescent actin molecules are incorporated at the barbed ends of filaments, while bleached actin is released from pointed ends. (**B**) Hippocampal neurons (DIV17) were transfected to coexpress GFP-actin and mCherry with or without LIMK1 shRNA, plus shr-LIMK1 rescue constructs as indicated. Dual color live images of individual dendrites are shown for each condition. (**C**) Images of baseline GFP-actin signal from the same dendritic regions shown in **B** (left column, t = 0 s, Pre-bleach). A Region of Interest (ROI, yellow circle) was photobleached and the dendrite was imaged immediately thereafter (second column images, 'Bleach') and at the indicated times post-bleach (third, fourth columns). (**D**) FRAP curves (normalized to average pre-bleach fluorescence), plotted from multiple single-spine ROIs for each condition from **C**. Values are mean ± SEM, n = 16–26 spines per condition. (**E**) Histogram of the stable fraction of GFP-actin (mean ± SEM) between t = 250 s and t = 300 s, calculated for each individual FRAP trace used to generate the pooled data in **D**. (Control vector: 8.3 ± 2.8%; LIMK1 knockdown: 32.2 ± 3.6%; shr-wt-LIMK1 'rescue': 15.3 ± 2.4%; shr-CCSS-LIMK1 'rescue': 41.0 ± 4.0%. *, p < 0.05 compared to control vector, n.s.; not significantly different, ANOVA with Dunnett's *post hoc correction*.

The following figure supplements are available for figure 4:

**Figure supplement 1**. Schematic and efficacy of shRNA knockdown/rescue approach.

**Figure supplement 2**. Further analysis of FRAP data confirms that acute loss of palmitoyl-LIMK1 increases the pool of stable GFP-actin in dendritic spines.

---

Mann–Whitney U test) but not by shr-CCSS-LIMK1 (interleaved control: 141 ± 10%; CCSS-LIMK1: 115 ± 4%, p = 0.031, Mann–Whitney U test; *Figure 5C–F*, *Figure 5—figure supplement 1*). These findings suggest that palmitoyl-LIMK1 is required for activity-dependent spine enlargement.

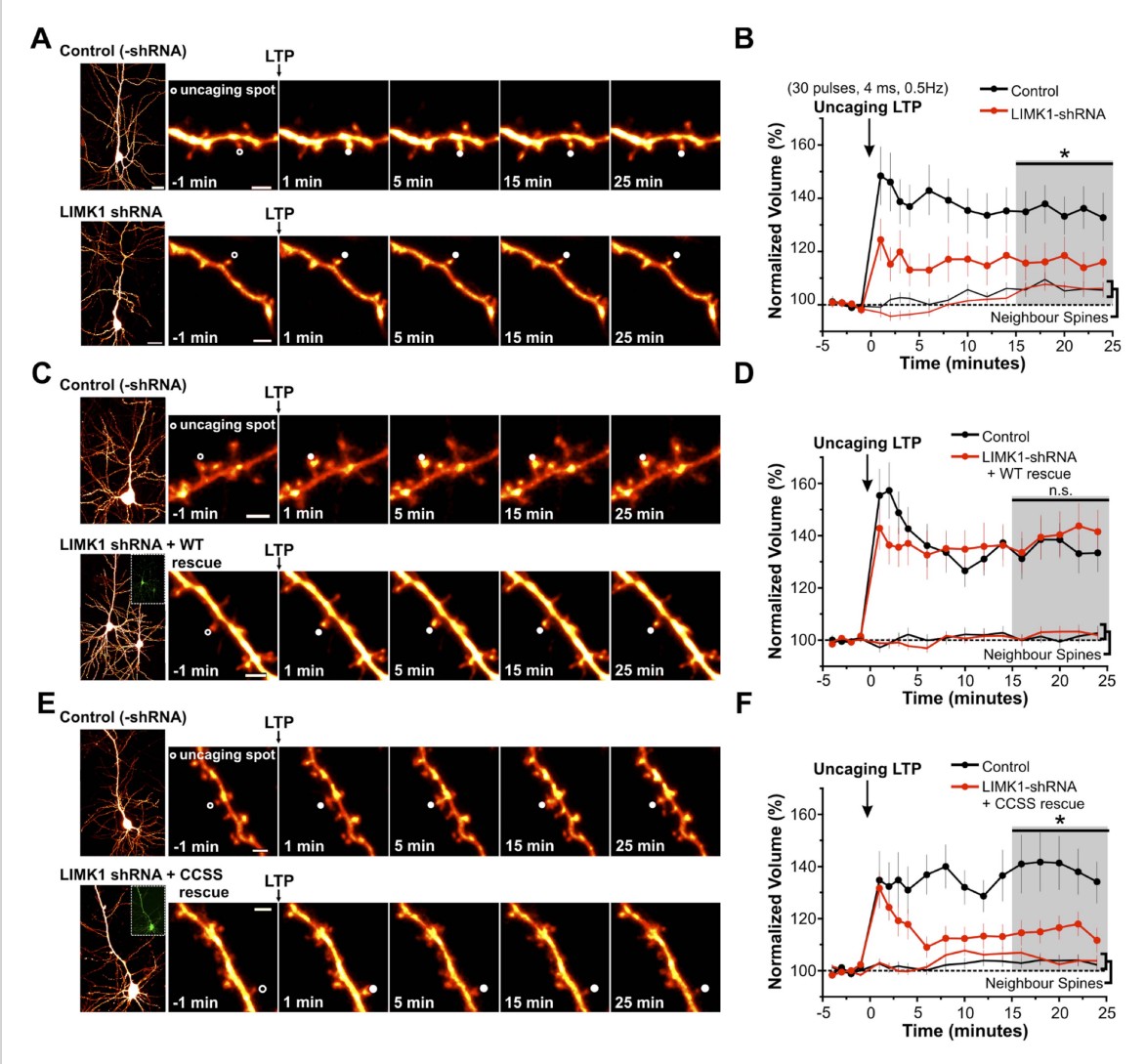

**Figure 5**. Palmitoyl-LIMK1 is required for spine-specific activity-dependent morphological plasticity. (**A**) *Left panels*: images of individual neurons in organotypic hippocampal slices expressing mCherry with or without LIMK1 shRNA. Magnified images of individual dendrites were acquired at the indicated times prior to and following uncaging of MNI-Glutamate on the head of the indicated spine (white circles). Scale bars: low magnification, 20 µm, magnified, 2 µm. (**B**) Time course of normalized spine-head volume (mean ± SEM) of stimulated and neighboring spines (control: 24 stimulated spines, 72 neighbors, 6 neurons; LIMK1 shRNA: 28 stimulated spines, 84 neighbors, 6 neurons). Average normalized spine volume (15–25 min post-uncaging) was plotted and used for statistical comparison (*: p < 0.05, Mann–Whitney U test, details in main text). (**C**, **D**) Images and time courses plotted as in **A**, **B** for neurons expressing mCherry (Control; 21 stimulated spines, 63 neighbors, 6 neurons) or mCherry, LIMK1 shRNA and shr-wt-LIMK1-GFP (LIMK1-shRNA + WT rescue; 27 stimulated spines, 81 neighbors, 8 neurons). (**E**, **F**) Images and time courses plotted as in **A**, **B** for neurons expressing mCherry (control; 23 stimulated spines, 69 neighbors, 6 neurons) or mCherry, LIMK1 shRNA and shr-CCSS-LIMK1-GFP (LIMK1-shRNA+ CCSS rescue; 28 stimulated spines, 84 neighbors, 8 neurons).

The following figure supplements are available for figure 5:

**Figure supplement 1**. Probability of spine growth or shrinkage in response to glutamate uncaging.

**Figure supplement 2**. 2-photon uncaging of MNI-glutamate induces spine-specific increases in both spine volume and synaptic strength.

## Long term loss of palmitoyl-LIMK1 causes spine and synapse elimination

Acute loss of palmitoyl-LIMK1 markedly impaired spine actin turnover without affecting dendritic spine number (*Figure 4*, *Figure 4—figure supplement 1*). However, we hypothesized that chronically

impaired actin turnover and activity-dependent plasticity caused by absence of palmitoyl-LIMK1 (*Figures 4, 5*), might detrimentally impact spine stability. To address this possibility, we transfected mature neurons to express GFP plus/minus LIMK1 shRNA and examined spine density 5 days later. This prolonged LIMK1 knockdown significantly reduced dendritic spine density, which was rescued by cotransfected shr-wt-LIMK1 but not by shr-CCSS-LIMK1 (*Figure 6A,B*). Absence of LIMK1 concomitantly reduced the number of excitatory synapses (defined as spines positive for pre- and postsynaptic markers; *Figure 6C,D*). Loss of both spines and synapses was rescued by shr-wt-LIMK1

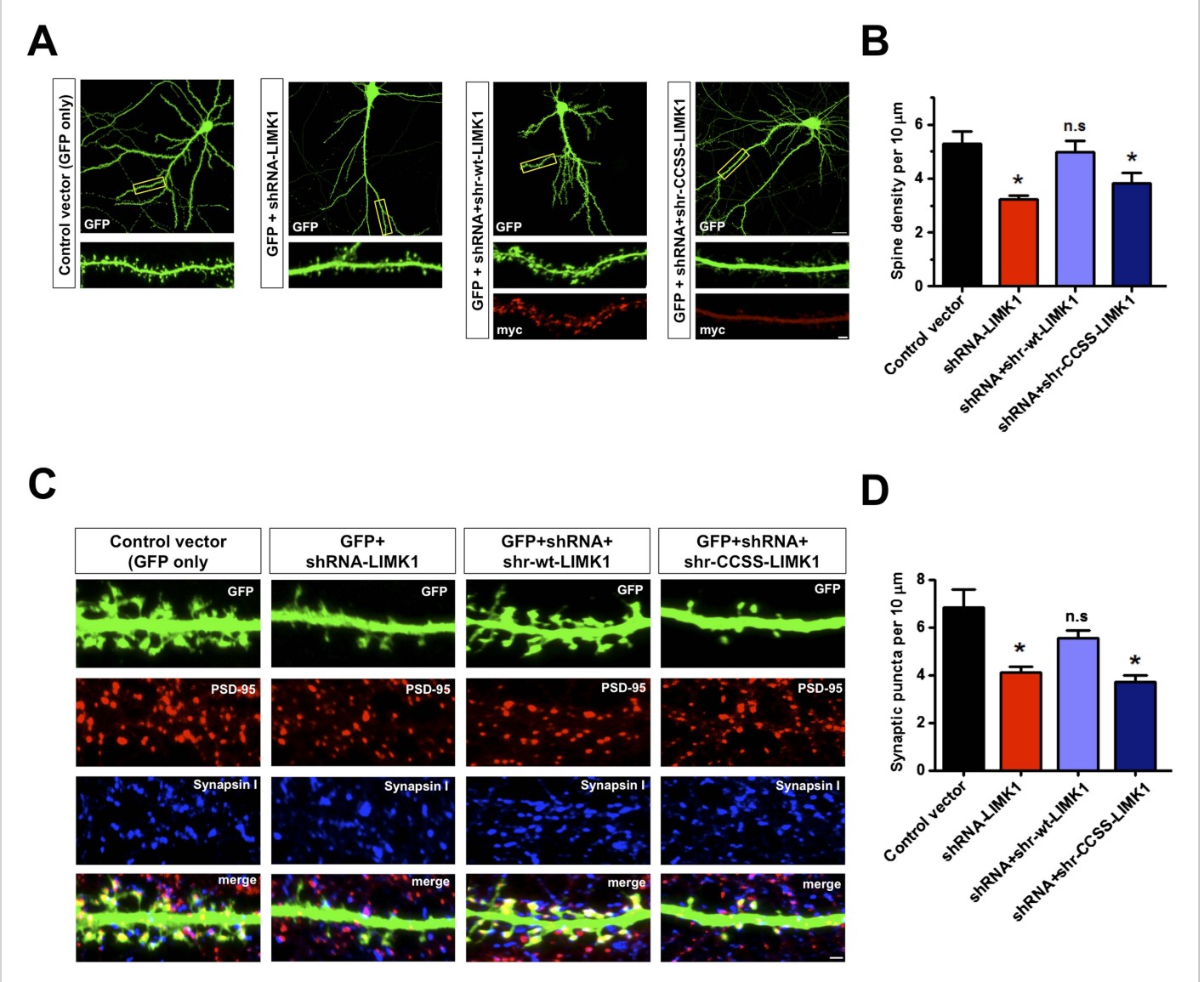

**Figure 6**. Prolonged loss of palmitoyl-LIMK1 reduces dendritic spine and synapse number. (**A**) Hippocampal neurons (DIV17) transfected to express GFP alone (first panel), GFP plus LIMK1 shRNA (second panel), or GFP plus LIMK1 shRNA, plus the indicated LIMK1 'rescue' constructs (third, fourth panels). Neurons were fixed 5 days later and immunostained to detect GFP and myc. Scale bar: 20 μm. Lower panels show magnified images of single dendrites (scale bar: 1 μm). (**B**) Spine density per 10 μm dendritic length from multiple neurons from **A** (mean ± SEM: control vector: 5.38 ± 0.48 spines; LIMK1 shRNA: 3.23 ± 0.15 spines; shRNA plus shr-wt-LIMK1: 4.99 ± 0.41 spines; shRNA plus shr-CCSS-LIMK1: 3.82 ± 0.40 spines; *p < 0.05 compared to control vector, ANOVA, Dunnett's *post hoc correction*. N = 40–50 neurons per condition). (**C**) Neurons transfected as in **A** were immunostained to detect GFP (morphology marker, green), presynaptic marker synapsin I (blue) and postsynaptic marker PSD-95 (red) (scale bar, 1 μm). (**D**) Quantified density (mean ± SEM) of colocalized PSD-95 and synapsin I puncta (morphologically defined synapses) per 10 μm dendritic length per condition from **C**. (Synaptic puncta: vector alone: 6.83 ± 0.76; LIMK1 shRNA: 4.12 ± 0.21; shRNA plus shr-wt-LIMK1: 5.54 ± 0.32; shRNA plus shr-CCSS-LIMK1: 3.71 ± 0.28; *p < 0.05 compared to control vector, ANOVA with Dunnett's *post hoc correction*. N = 30 neurons per condition).

but not shr-CCSS-LIMK1 (*Figure 6C,D*). Together, these results suggest that prolonged loss of palmitoyl-LIMK1 leads to spine instability and synapse loss.

## Dual palmitoylation is critical for neuronal LIMK1 activation via a CaMKII/PAK-dependent pathway

The failure of CCSS-LIMK1 to rescue effects of LIMK1 knockdown to control spine actin turnover, morphological plasticity and spine stability (*Figures 4–6*) was striking because CCSS-LIMK1 is not absent from spines, only not enriched in spine heads (*Figure 1*). Indeed, despite robust expression, CCSS-LIMK1 is essentially a null mutant (*Figures 4–6*), suggesting that dual palmitoylation controls not only LIMK1 localization, but also LIMK1 function in neurons.

To test this hypothesis, we first addressed whether CCSS mutation affects LIMK1 function in vitro. In in vitro kinase assays, LIMK1's upstream activator PAK3 phosphorylated wt-LIMK1 and CCSS-LIMK1 with similar kinetics and to a similar extent (assessed by a phospho-specific antibody recognizing LIMK1's activation site (T508); *Figure 7A,B*). Moreover, following activation by PAK3, wt- and

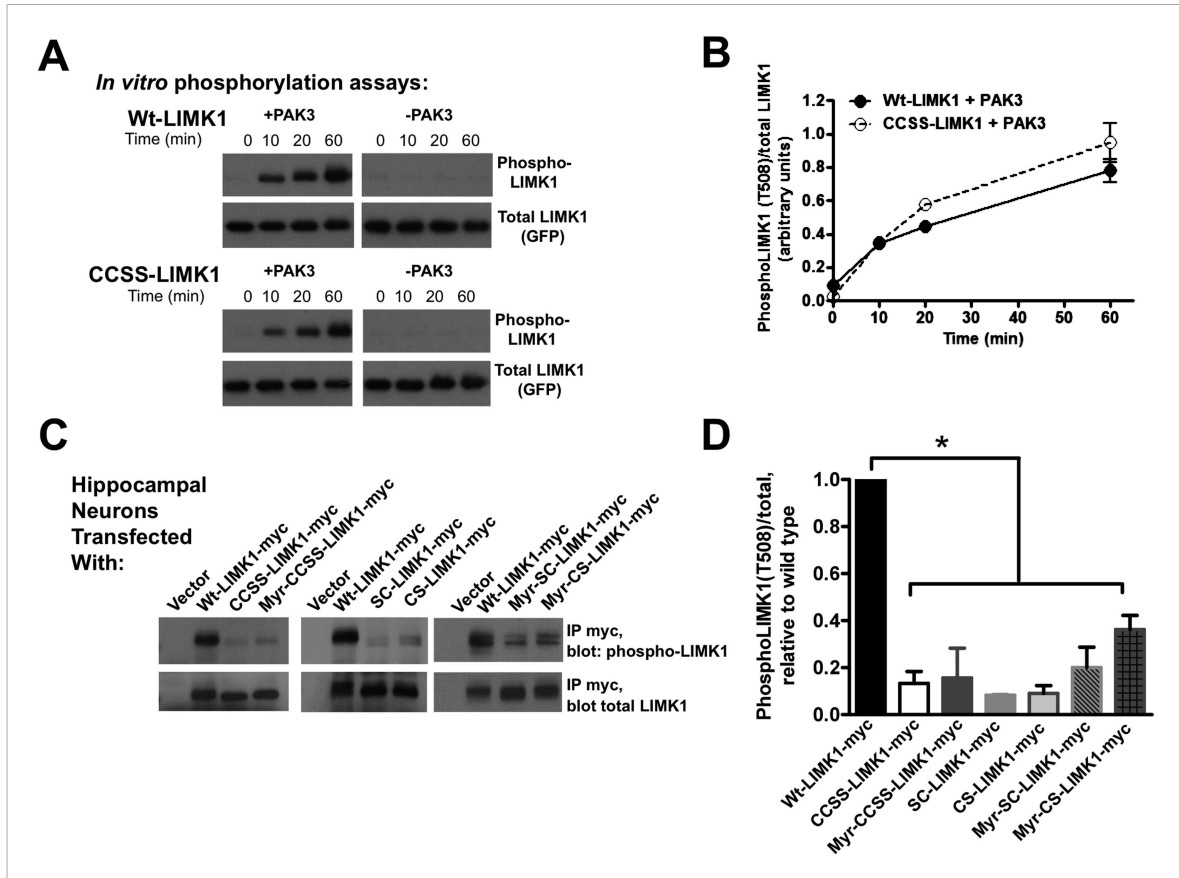

**Figure 7**. Di-palmitoylation is critical for LIMK1 activation in neurons. (**A**) CCSS mutation does not affect LIMK1 T508 phosphorylation in vitro. Wt- and CCSS-LIMK1-GFP were incubated in vitro with Mg-ATP, with or without active PAK3. Reactions were stopped at the indicated times, subjected to SDS-PAGE and immunoblotted to detect phospho-T508 LIMK1 and total LIMK1. (**B**) Quantified data (mean ± SEM, N = 3 determinations per condition) from assays in **A**. Timecourse and extent of phosphorylation of Wt- and CCSS-LIMK1 by PAK3 is similar in vitro. (**C**) Dual palmitoylation is uniquely required for LIMK1 T508 phosphorylation in neurons. Hippocampal neurons transfected with the indicated myc-tagged LIMK1 constructs were lysed and myc immunoprecipitates were immunoblotted to detect phospho-T508 and total LIMK1. (**D**) Quantified signals (mean ± SEM) for N = 3–6 determinations per condition from **C**. *p < 0.05 compared to wt-LIMK1-myc. ANOVA with Dunnett's *post hoc correction*. Note that phosphorylation of Myr-CS LIMK1 differs from that of CS-LIMK1 or Myr-CCSS-LIMK1 (ANOVA). This result suggests that, at least for Myr-CS-LIMK1, addition of the myristolyation tag does not interfere with recognition of the remaining palmitoyl-site.

The following figure supplement is available for figure 7:

**Figure supplement 1**. CCSS mutation does not affect phosphorylation of cofilin by LIMK1.

CCSS-LIMK1 phosphorylated their downstream substrate cofilin to a similar extent in vitro (*Figure 7—figure supplement 1*). These results suggest that CCSS mutation affects neither LIMK1's phosphorylation by PAK3, nor LIMK1's ability to phosphorylate cofilin. It is thus also unlikely that CCSS mutation grossly affects LIMK1 structure.

In striking contrast, T508 phosphorylation of wt-LIMK1 in hippocampal neurons was approximately 10-fold higher than that of CCSS-LIMK1. SC-LIMK1, CS-LIMK1 and Myr-CCSS-LIMK1 mutants were also only weakly phosphorylated in neurons (*Figure 7C,D*). The doubly lipid-modified mutants, in particular Myr-CS-LIMK1, were phosphorylated to a slightly greater extent than CCSS-LIMK1, but still significantly less than wild type LIMK1 (*Figure 7C,D*). These results suggest that spine targeting (which is only achieved by dual palmitoylation), and not membrane association per se, is the key factor that controls LIMK1 phosphorylation and activation in neurons.

These findings suggest that a signaling pathway assembled on the spine membrane activates palmitoyl-LIMK1. However, despite links between LIMK1 and activity-dependent spine signaling (*Bosch et al., 2014*), the pathway that activates LIMK1 in spines is unclear. LIMK1 can be phosphorylated by either PAK or ROCK family kinases (*Edwards et al., 1999*; *Maekawa et al., 1999*), both of which are active at the membrane because they bind active forms of Rac/Rho/Cdc42 small G proteins. Interestingly, PAK inhibition, but not ROCK inhibition, greatly reduced LIMK1 phosphorylation in hippocampal neurons (*Figure 8A,B*), suggesting that PAK is the key LIMK1 activator in spines. This conclusion is supported by the robust phosphorylation of LIMK1 by PAK3 in vitro (*Figure 7A*), and by prior links between PAKs and activity-dependent morphological plasticity of single spines (*Murakoshi et al., 2011*). Moreover, consistent with findings that spine morphological plasticity requires CaMKII (*Lee et al., 2009*), a CaMKII inhibitor also greatly reduced LIMK1 phosphorylation (*Figure 8A,B*). Together, these findings suggest that CaMKII in spines triggers Rac/Cdc42 activation, which recruits active PAK to phosphorylate and activate wtLIMK1 specifically on the spine membrane.

## Discussion

### Palmitoylation of LIMK1 controls actin dynamics in dendritic spines

The importance of palmitoylation in neuronal regulation is increasingly appreciated, but most studies have focused on how this modification controls the localization of receptors and their 'scaffold' protein partners (*Fukata and Fukata, 2010*; *Thomas and Huganir, 2013*). This study reveals a novel role for palmitoylation in the spatial control of signaling events that regulate neuronal actin polymerization. It has been unclear how actin regulatory proteins, many of which are predicted to be

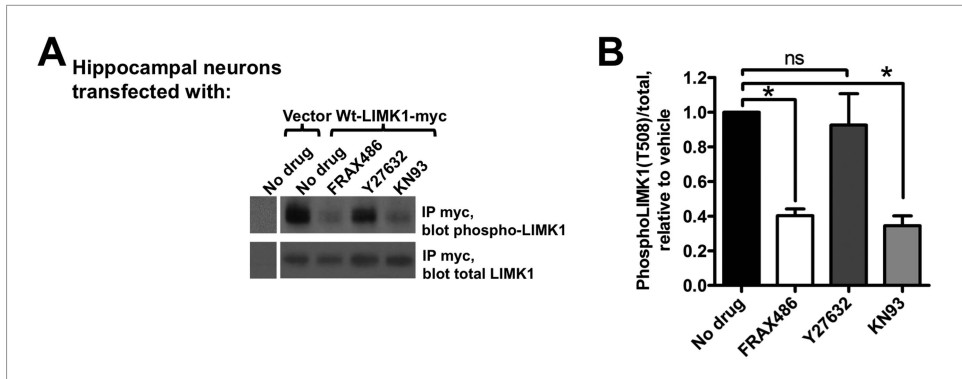

**Figure 8**. Activation of palmitoyl-LIMK1 in neurons requires CaMKII and PAK, but not ROCK. (**A**) Hippocampal neurons transfected with the indicated cDNAs were left untreated, or were incubated for 1 hr with PAK inhibitor FRAX 486 (5 μM), ROCK inhibitor Y27632 (10 μM), or CaMKII inhibitor KN93 (10 μM) prior to lysis. Myc immunoprecipitates were blotted with the indicated antibodies. (**B**) Quantified data of multiple determinations from **A** confirms that LIMK1 phosphorylation in neurons is PAK- and CaMKII-dependent, but not ROCK-dependent. (*p < 0.05 compared to control (no drug), ns: not significant. ANOVA with Dunnett's *post hoc* correction. n = 5–7 determinations per condition).

diffusible, can control spatially precise changes in dendritic spine morphology. Our results suggest that one key factor in the spatial control of actin dynamics is palmitoylation of the actin regulator LIMK1, which is required for normal actin turnover in spines, for spine-specific structural plasticity and for long-term spine stability. These findings reveal new mechanisms that may govern both spine-specific morphological plasticity and perhaps also intra-spine control of actin dynamics.

## How does palmitoylation regulate LIMK1 localization and signaling?

The functional requirement for palmitoyl-LIMK1 in spine-specific regulation likely arises from at least three different effects of palmitoylation on LIMK1 at the cellular and molecular level. First, palmitoylation is critical for LIMK1 enrichment in dendritic spines (*Figure 1*). Interestingly, even LIMK1 mutants that are predicted to be dually lipidated (Myr-SC- and Myr-CS-LIMK1) are not spine-enriched, suggesting that dual palmitoylation is the key factor that controls LIMK1 spine targeting. Consistent with this notion, LIMK1's dual palmitoylation motif is sufficient to target a heterologous protein (GFP) to spines (*Figure 2*). It is remarkable that such specific targeting information is contained within such a short sequence. Moreover, to our knowledge, no other specific spine-targeting motif has been described. LIMK1's 1–15 sequence could hence be extremely useful to deliver any protein of interest specifically to spines, which could facilitate a range of studies into spine and synaptic biology.

Second, palmitoylation increases the stable fraction of LIMK1 in spines and thus helps to anchor LIMK1 in spine heads (*Figure 1—figure supplement 1*). However, although this effect is significant, even CCSS-LIMK1 fluorescence recovery is far slower than that reported for cytosolic GFP (*Star et al., 2002*; *Zheng et al., 2010*) and is also markedly slower than that of the proteasome subunit Rpt1, a protein of similar size to LIMK1 (*Bingol et al., 2010*). Other factors, most likely protein–protein interactions, are thus likely major determinants of LIMK1's limited diffusibility in spines, with palmitoylation exerting an additional stabilizing effect.

Third, dual palmitoylation is critical for LIMK1 phosphorylation at its activatory T508 site in neurons. In contrast, palmitoyl-site mutants (CCSS-LIMK1, SC-LIMK1 and CS-LIMK1), although not absent from spines, are not phosphorylated, and phosphorylation of myristoylated LIMK1 mutants (Myr-CCSS-LIMK1, Myr-SC-LIMK1, Myr-CS-LIMK1) is also very low. These results suggest that enrichment on the spine membrane, rather than membrane attachment per se, is essential for LIMK1 phosphorylation in neurons. A plausible reason for this spine-specific activation is that key LIMK1 activators, in particular active forms of CaMKII and PAK, are also specifically enriched on or adjacent to the spine membrane ([*Shen and Meyer, 1999*; *Zhang et al., 2005*], *Figure 9*).

## Cdc42/Rac, PAK and palmitoyl-LIMK1 are critical for activity-dependent plasticity of single spines

The palmitoylation-dependence of LIMK1 localization and activation sheds new light on mechanisms that spatially restrict signaling to selected spines. The small G proteins Cdc42 and Rac are critical for activity-dependent spine enlargement (*Murakoshi et al., 2011*), but their ability to ensure spatially precise actin regulation would appear limited if key 'downstream' effectors such as PAKs and LIMKs (*Edwards et al., 1999*; *Murakoshi et al., 2011*) were freely diffusible. However, because synaptic localization of LIMK1 is tightly regulated by palmitoylation (*Figure 1*), Rac/Cdc42/PAK/LIMK1 signals can remain spatially localized, ensuring spine-specific actin regulation (*Figure 9*).

Interestingly, molecular requirements for spine-specific morphological plasticity and changes in glutamate-evoked transmission (i.e., activity-dependent functional plasticity) are very similar, with both requiring both Cdc42 signaling and also actin polymerization (*Matsuzaki et al., 2004*; *Murakoshi et al., 2011*). We found that very similar stimulation protocols trigger changes in spine volume that are palmitoyl-LIMK1-dependent (*Figure 5*) and also induce spine-specific functional plasticity (*Figure 5—figure supplement 2*). These findings raise the possibility that palmitoyl-LIMK1 is essential not only for structural but also functional plasticity, and may be a key link between the spine-specific Cdc42 signaling and actin polymerization described by others.

## Downstream targets of palmitoyl-LIMK1 that control actin dynamics

Our study provides new insights into spatial control of the dendritic spine cytoskeleton, but what substrate(s) is responsible for palmitoyl-LIMK1-dependent effects on dendritic spine actin turnover (*Figure 4*) and activity-dependent spine enlargement (*Figure 5*)? Multiple lines of evidence suggest

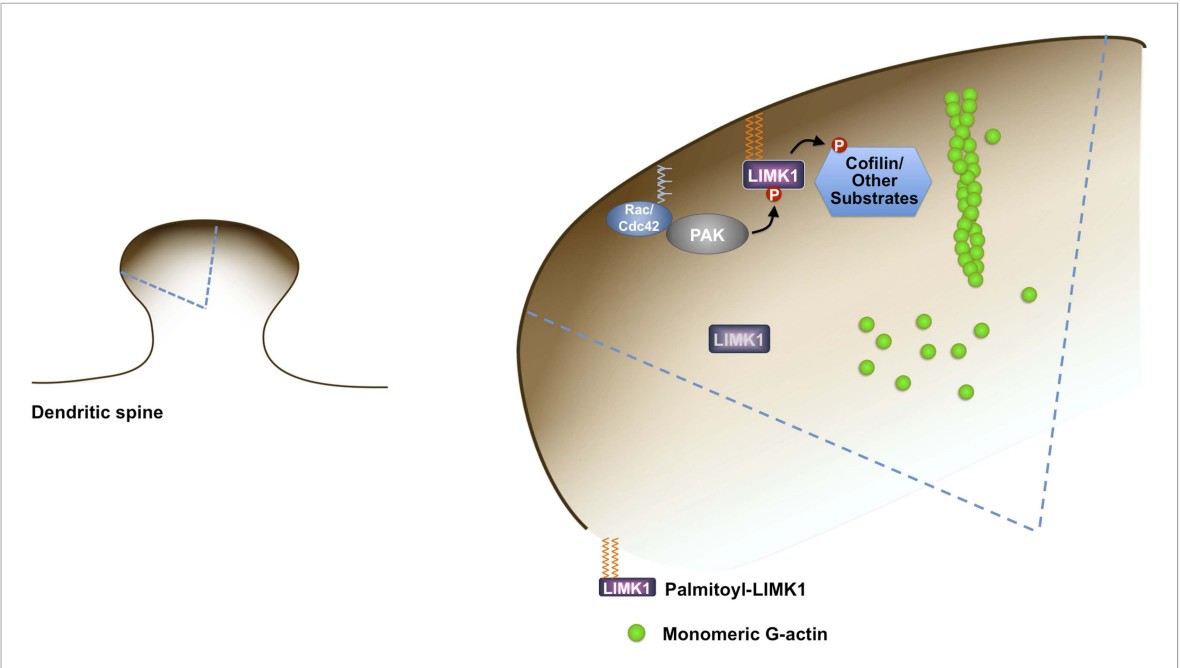

**Figure 9**. Model of palmitoyl-LIMK1-dependent control of actin dynamics in dendritic spines. Dendritic spine schematic (left panel) with the indicated region expanded (blue boundary) to the right. Palmitoylation targets LIMK1 to the spine membrane, where it is phosphorylated by membrane-bound activators such as Rac/Cdc42/PAK. In contrast, any non-palmitoylated LIMK1 in the spine 'core' remains inactive. By governing both LIMK1 localization and activation, palmitoylation may also facilitate local, juxtamembrane-specific phosphorylation of cofilin and/or other LIMK1 substrates to enhance the spatial control of actin filament turnover in spines. For clarity, other actin regulatory processes such as filament capping and branching are not shown.

a key role for cofilin, LIMK1's best known substrate, in these processes. Not only is cofilin phosphorylation critical for activity-dependent spine enlargement (*Gu et al., 2010*), but elevated neuronal activity rapidly increases both spine volume (*Figure 5*; [*Matsuzaki et al., 2004*; *Yang et al., 2008*]) and also endogenous levels of phosphorylated, inactive cofilin (*Rex et al., 2009*). However, despite these links, it is possible that additional and/or different palmitoyl-LIMK1 substrates contribute to the functional effects that we observe.

Even if cofilin is the key substrate via which palmitoyl-LIMK1 exerts effects on spines, some of our results are worthy of further discussion. One might predict that increases in dephospho- (active) cofilin, caused by LIMK1 knockdown, would lead to an excess of short actin filaments and/or actin monomers, so that GFP-actin fluorescence in our FRAP experiments would recover more rapidly and/or to a greater extent. Surprisingly, though, a significant percentage of actin in LIMK1 knockdown (and CCSS-LIMK1 'rescue') spines appears to be very stable. One possible explanation for this finding is that LIMK1 knockdown may increase the formation of cofilin-actin rods (*Bamburg et al., 2010*), stable structures that are formed only by dephospho-cofilin (*Bernstein et al., 2006*). However, while well documented within neurites, cofilin-actin rods have not been reported within spines.

A second explanation arises from the ability of active (dephospho-) cofilin to induce a twist in actin filament conformation that propagates along the filament (*Bamburg et al., 1999*). Filaments are more likely to sever at boundaries between cofilin-bound (twisted) and unbound (untwisted) regions (*Bobkov et al., 2006*). However, an entire cofilin-bound filament, as might occur in some LIMK1 knockdown (or CCSS-LIMK1 rescue) spines, is more stable than a bare filament (*Dedova et al., 2004*; *Andrianantoandro and Pollard, 2006*). Thus, LIMK1 knockdown may lead to a subset of filaments that are decorated with active cofilin but which are actually more stable.

A third, intriguing explanation arises from findings that LTD-like stimuli (low frequency electrical stimulation or bath application of NMDA) markedly increase actin filament stability in spines (*Star et al., 2002*). Strikingly, our LTP-like glutamate uncaging stimulus results in shrinkage of a subset of spines (an LTD-like effect), almost exclusively restricted to the LIMK1 knockdown and CCSS-LIMK1

rescue conditions (*Figure 5—figure supplement 1*). This suggests that some LIMK1 knockdown (and CCSS-LIMK1 rescue) spines respond to LTP-like inputs with LTD-like outputs that is, that their learning rules are impaired. LIMK1 knockdown and CCSS-LIMK1 'rescue' spines may therefore interpret spontaneous bursts of activity in cultured neurons as an LTD-like stimulus. In the short term this may result in decreased actin filament turnover, as previously reported (*Star et al., 2002*), while over longer times this impaired actin turnover may underlie the spine shrinkage and synapse loss seen in the absence of palmitoyl-LIMK1.

Finally, we cannot exclude the possibility that additional palmitoyl-LIMK1 substrate(s) rather than cofilin are responsible for effects on actin turnover and/or morphological plasticity. Super-resolution and/or electron microscopy approaches may provide more insight into how actin filament structure and/or turnover is altered in the absence of palmitoyl-LIMK1. However, while the molecular explanation remains to be elucidated, our results strongly suggest that palmitoylation of LIMK1 is critical for normal actin filament turnover in spines.

## Could palmitoyl-LIMK1 control polarized 'shell-to-core' actin regulation in spines?

If cofilin is indeed the key palmitoyl-LIMK1 substrate responsible for the functional effects that we observe then could this provide insight into the sub-spine control of actin dynamics? Previous studies reveal a directional flow of actin from the spine tip to the neck (*Honkura et al., 2008*; *Frost et al., 2010*). These findings are consistent with models of actin array treadmilling, in which new actin molecules are incorporated at actin filament barbed ends, predominantly located close to the plasma membrane (*Le Clainche and Carlier, 2008*; *Hotulainen et al., 2009*; *Bugyi and Carlier, 2010*). These models suggest that actin polymerization is favored in this juxtamembrane region, or 'shell' of the spine. Conversely, though, disassembly/severing must be favored towards the center of the spine head ('core'), both to prevent filament overgrowth and to supply new actin monomers for ongoing treadmilling.

Many proteins likely contribute to the intra-spine balance of actin filament polymerization/depolymerization. However, a key generator of actin monomers is cofilin, which enhances filament disassembly by severing 'aged' regions of filaments (those in which ADP-actin subunits predominate, usually toward filament pointed ends [*Mizuno, 2013*]). It has been hypothesized that in order for treadmilling to occur normally, LIMK1 may need to phosphorylate and inhibit cofilin specifically in juxtamembrane regions (*Ridley, 2006*). How such spatially precise cofilin regulation might be achieved has been unclear, but could be aided by the enrichment of active, palmitoylated LIMK1 on the spine membrane. Juxtamembrane-specific phosphorylation of cofilin by palmitoyl-LIMK1 would be predicted to locally favor actin filament polymerization to maintain or, in response to local synaptic cues, rapidly expand the size of the spine head. Conversely, palmitoyl-LIMK1's absence from the spine core would allow cofilin to remain active and disassemble/sever actin filaments, ensuring a supply of actin monomers for treadmilling and limiting excess filament polymerization. A key 'security feature' of this model is that inappropriately localized (depalmitoylated) LIMK1 is inactive in neurons and thus would not adversely affect such spatially precise cofilin regulation.

In support of this model, the phosphatase Slingshot-1L, which dephosphorylates and activates cofilin, is active only when bound to actin filaments (*Nagata-Ohashi et al., 2004*; *Soosairajah et al., 2005*) and would thus be predicted to be less active in juxtamembrane regions. However, we emphasize that such a cofilin gradient model is speculative, and that overall cofilin activity in spines is likely controlled not only by LIMK1, but also by proteins such as Actin-interacting protein-1 (Aip1) that regulate cofilin activity, and those such as Coronins that control cofilin-actin binding (*Ono, 2003*; *Cai et al., 2007*; *Kueh et al., 2008*). Nonetheless, an intriguing possibility arising from this model is that palmitoylation of signaling enzymes (discussed further below) is used by neurons to establish or maintain polarized (shell-to-core, or edge-to-center) signaling gradients.

## What lies upstream of LIMK1 palmitoylation?

Another key question regards how LIMK1 palmitoylation is regulated in spines. Changes in neuronal activity acutely alter palmitoylation of a subset of synaptodendritic proteins (*Kang et al., 2008*). However, our preliminary findings suggest that LIMK1 does not fall into this category, because LIMK1 palmitoylation in ABE assays is unaltered by treatment of neurons with Bicuculline or KCl (JG and GT, unpublished observations). However, it is still possible that extracellular stimuli, particularly those such as ephrins that acutely regulate spine dynamics via PAK (*Penzes et al., 2003*), may dynamically alter

LIMK1 palmitoylation. In addition, limitations of the ABE assay (which detects the entire cellular complement of palmitoyl-LIMK1, irrespective of location) could mask selective changes in LIMK1 palmitoylation at spines.

It is also unclear which PAT(s) controls LIMK1 palmitoylation in spines. Multiple PATs can palmitoylate LIMK1 in cotransfected HEK293T cells, including both Golgi-localized and synaptoden-dritic PATs (JG and GT, unpublished observations). This finding suggests that no single PAT is likely to control LIMK1 palmitoylation. It is also unclear whether the activity of LIMK1's upstream activators PAK and/or CaMKII is required for LIMK1 palmitoylation. These are all interesting questions to address in the future.

### New roles for palmitoylation in the spatial control of neuronal signaling

Two broader points that emerge from our findings is that palmitoylation can ensure spatially precise control of actin regulation and of kinase signaling. With this in mind, it is interesting that the actin regulators profilin and coronin are also likely palmitoylated (*Kang et al., 2008*), suggesting that palmitoylation may control multiple aspects of spine-specific actin regulation.

In addition, recent reports (*Takemoto-Kimura et al., 2007*; *Yang et al., 2012*) and our own ongoing experiments (in which we have identified several other palmitoyl-kinases; JG and GT, unpublished observations) suggest that signaling by other kinases is palmitoylation-dependent. Moreover, numerous other signaling enzymes contain predicted palmitoylation sites similar to those found in LIMK1 (not shown). Neurons may thus broadly use palmitoylation to localize diverse groups of signaling enzymes to enhance the spatial specificity of myriad intracellular signaling events.

### Links between LIMK1 palmitoylation, cytoskeletal regulation and cognitive function

Aberrant regulation of the spine actin cytoskeleton is strongly linked to impaired cognition. There has been considerable interest in LIMK1 in this regard, due to *LIMK1*'s frequent genetic deletion in Williams syndrome (*Frangiskakis et al., 1996*; *Tassabehji et al., 1996*) and the impaired performance of *LIMK1* knockout mice in learning tasks (*Meng et al., 2002*). This latter phenotype was linked to impaired spine morphology, but whether LIMK1 acts directly in spines was unclear. Our results strongly support this hypothesis and further implicate LIMK1 palmitoylation as critical for the control of spine structure. Moreover, differential spine targeting of LIMK1 and LIMK2 (*Figure 3*) may explain LIMK2's failure to compensate following loss or mutation of LIMK1.

We do, however, note some differences between effects of germline *LIMK1* knockout (*Meng et al., 2002*) and LIMK1 knockdown in mature neurons (this study). In particular, conventional *LIMK1* knockout mainly affects spine morphology and PSD size, while LIMK1 knockdown reduces spine and synapse numbers. The more subtle effects of conventional *LIMK1* knockout may be due to developmental compensation by other pathways. Indeed, acute knockdown of other synaptic proteins causes more dramatic effects than germline knockout (*Elias et al., 2006*). Nonetheless, our findings strengthen links between LIMK1 and the control of spine morphology and plasticity, both of which are linked to higher brain function.

Finally, spine-associated impairments in the absence of palmitoyl-LIMK1 are reminiscent of phenotypes seen in human patients with Intellectual Disability and other cognitive disorders (*Fiala et al., 2002*; *Nadif Kasri and Van Aelst, 2008*). Moreover, mutations in Rac/Cdc42 regulators, PAKs and LIMK1 are linked to disrupted spine morphology and impaired cognition (*Frangiskakis et al., 1996*; *Tassabehji et al., 1996*; *Allen et al., 1998*; *Nadif Kasri and Van Aelst, 2008*). Our findings provide further evidence that a broadly similar mechanism (i.e., impaired cytoskeletal regulation in spines) may underlie many of these conditions.

## Materials and methods

### Antibodies

The following antibodies, from the indicated sources, were used: purified myc 9E10 (Enzo Life Sciences, Farmingdale, NY); PSD-95 (K28/43) (Neuromab, UC Davis, CA); Synapsin I, chicken anti-GFP (EMD Millipore, Billerica, MA); mouse anti-GFP 3E6 (Life Technologies, Carlsbad, CA); LIMK1 (BD Biosciences, San Jose, CA); myc 9B11, LIMK2 8C11, PhosphoLIMK1(T508)/LIMK2(T505), cofilin, phospho-cofilin (Cell Signaling Tech, Danvers, MA); Tubulin (Sigma, St. Louis, MO).

## Chemicals

2-Bromopalmitate and S-Methyl methanethiosulfonate (MMTS) were from Sigma. All other chemicals were from ThermoFisher Scientific (Waltham, MA) and were of the highest reagent grade.

## Molecular biology and cDNA clones

Human LIMK1 and LIMK2 cDNAs were from Arizona State University plasmid repository. A C-terminal myc tag was added to LIMK1 and LIMK2 by PCR and the resultant fragments were subcloned into the mammalian expression vector pRK5 and the lentiviral vector FUW (*Lois et al., 2002*). CCSS-LIMK1-myc (Cys 7, 8 of LIMK1 mutated to Ser), SC-LIMK1-myc and CS-LIMK1-myc were generated by PCR using mutagenic primers. An N-terminal myristoylation sequence (MGQSLTT; [*Wyszynski et al., 2002*]) was added to the N-terminus of CCSS-, CS and SC LIMK1 mutants by PCR. A LIMK1 shRNA (GAACGTGGTGGTGGCTGAC) was subcloned into a modified FUGW vector (*Lois et al., 2002*) downstream of an H1 promoter, and its effectiveness was confirmed against cotransfected rat LIMK1-myc cDNA (purchased from Origene, Rockville, MD). The GFP cassette of FUGW was removed and mCherry cDNA inserted to generate FUmChW. ShRNA resistant (shr) wt- and CCSS- LIMK1-myc were generated by mutating the shRNA target region while maintaining protein-coding sequence. Shr-wt- and CCSS-LIMK1-GFP were generated by PCR amplification of shr-wt- and shr-CCSS-LIMK1 cDNAs, without the myc tag and subsequent ligation into eGFP-N2 vector (Clontech, Mountain View, CA).

## Bioinformatic identification of LIMK1 as a predicted palmitoyl-protein

Two bioinformatic approaches were used to identify palmitoylated actin regulators. First, we searched for known actin regulators among the top 'hits' in a database of predicted palmitoyl-proteins, originally generated using the CSS-Palm prediction program (*Ren et al., 2008*). Second, we used the Regular Expression function of Scansite (*Obenauer et al., 2003*) to identify known actin regulatory proteins that contain CXC or CC motifs (C: cysteine; X: any amino acid) within their first 10 residues, as motifs of this type are frequently palmitoylated (*Fukata and Fukata, 2010*).

## Cultured hippocampal neurons

Hippocampi were dissected from E18 rat embryos and neurons were cultured in Neurobasal/B27 as described (*Thomas et al., 2012*). All animal use protocols were approved by the Institutional Animal Care and Use Committee of Temple University.

## Acyl Biotinyl exchange assay (ABE)

ABE was performed essentially as described (*Thomas et al., 2012*). For ABE experiments, transfected HEK293T cells or neurons were lysed directly in buffer containing 50 mM HEPES pH 7.0, 2% SDS, 1 mM EDTA plus protease inhibitor cocktail (PIC, Roche, Indianapolis, IN) and 20 mM methyl-methane thiosulfonate (MMTS, to block free thiols). Excess MMTS was removed by acetone precipitation and pellets were resuspended in buffer containing 4% (wt/vol) SDS. BCA assays (Life technologies, Grand Island, NY) were performed to normalize total protein amounts when samples from different developmental stages were compared. Samples were diluted and incubated for 1 hr at room temperature in either 0.7 M hydroxylamine pH 7.4 (to cleave thioester bonds) or 50 mM Tris pH 7.4. Acetone precipitation was performed to remove hydroxylamine or Tris. Pellets were resuspended in 4% (wt/vol) SDS, diluted in 50 mM Tris pH 7.4 containing sulfhydryl-reactive (HPDP-) biotin and incubated for 1 hr at room temperature. Unreacted HPDP-biotin was removed by acetone precipitation and pellets were resuspended in lysis buffer without MMTS. Samples were diluted to 0.1% (wt/vol) SDS and biotinylated proteins were affinity-purified using neutravidin-conjugated beads. Beta-mercaptoethanol (1% [vol/vol]) was used to cleave HPDP-biotin and release biotinylated proteins from the beads. The released proteins in the supernatant were denatured in SDS sample buffer and processed for Western blotting with LIMK1 antibodyABE assays and all other biochemical experiments were performed at least 3 times. In each case a representative experiment is shown.

## Lentiviral infection and shRNA knockdown

VSV-G pseudotyped lentivirus was produced in HEK293T cells as described (*Thomas et al., 2012*). Briefly, HEK293T cells were cotransfected with FUGW or FUmChW vectors (with or without

LIMK1shRNA) plus VSV-G, pMDLg and RSV-Rev helper plasmids. Supernatant containing virus was harvested 48 and 72 hr post-transfection, concentrated by ultracentrifugation, resuspended in Neurobasal medium and used to infect neurons at DIV9. Neurons were lysed at DIV15.

## Pharmacological treatments

2-Bromopalmitate was prepared as a 100 mM stock in ethanol and used at 100 µM final concentration. Sister cultures were treated with solvent control (0.1% [vol/vol] ethanol).

## Transfection and immunocytochemistry

HEK293T cells were transfected as described (*Thomas et al., 2005*). Hippocampal neurons on coverslips were transfected at DIV13-18 using Lipofectamine 2000 (Invitrogen) as described (*Thomas et al., 2012*). For immunocytochemistry, neurons were fixed in 4% (wt/vol) paraformalde-hyde, 4% (wt/vol) sucrose in phosphate-buffered saline (PBS), washed with PBS and permeabilized with PBS containing 0.25% (wt/vol) Triton X-100. After brief PBS washes, coverslips were blocked for 30 min at room temperature in 10% (vol/vol) normal goat serum (NGS) diluted in PBS, incubated overnight at 4°C with primary antibodies (in NGS/PBS) and then with AlexaFluor-conjugated secondary antibodies for 1 hr at room temperature. For localization experiments, neurons were always fixed <24 hr post-transfection.

## Image acquisition and analysis of dendritic spine morphology and synaptic puncta

For confocal imaging of fixed neurons, Z-stack images (0.2 µm spacing, 1024 × 1024 pixel resolution) were acquired using a Nikon C2 inverted confocal microscope with a 60× oil immersion objective (1.4 NA, plan-Apo). Acquisition parameters (laser power, gain and offset) were kept constant between all conditions. Maximum intensity projections were generated using NIS Elements software and used for mask analysis of mean intensities for spine to shaft intensity ratio and colocalization for synaptic puncta.

## Analysis of spine targeting ratio, intensity profiles and spine morphology

Quantitative analysis of fluorescent intensity line profiles and spine to shaft ratio of confocal stacked images were performed using Nikon NIS-Elements AR software. To quantify spine to shaft targeting ratios, dendritic spine were identified in the GFP channel. The fluorescent intensities of myc signals in spine heads and directly adjacent shaft regions were manually measured using intensity profile tool (crossed lines) of NIS-Elements. Soma intensity profiles were constant in all images measured. Average intensities of spines and adjacent shaft regions were exported to Excel, ratios were calculated and data were plotted using Graphpad Prism. To determine the morphologies of dendritic spines, the outline of the dendrite, including all spines, was manually traced using the signal from a morphology marker (usually GFP). Area, length and head width of each spine was then measured by Metamorph software (Molecular Devices).

## FRAP of GFP-actin

Hippocampal neurons (DIV17) were transfected with GFP-β actin plus mCherry cDNAs as above. 24 hr later, neurons were transferred to a live imaging chamber (Warner Instruments) in recording buffer (*Thomas et al., 2012*), containing (in mM) HEPES 25, NaCl 120, KCl 5, CaCl2 2, Glucose 30 and MgCl2 1 (pH 7.4). The chamber was assembled on the Nikon C2 confocal microscope stage. Neurons were identified based on the cotransfected morphology marker mCherry. Images of dendritic segments were then acquired for both mCherry and GFP-actin signals, using 10× optical zoom with acquisition settings of 256 × 256 pixel resolution at 2% laser power. After acquiring images from both channels, FRAP was performed only on the GFP-actin channel. Prebleach fluorescent signal was acquired using a 488 nm line argon laser and recorded using a 500–550 nm band pass filter. A circular Region of Interest (ROI, 2 µm diameter) on a selected dendritic spine head was photobleached by scanning with the 488 nm argon laser line at 100% laser power with pixel dwell time of 2.2 µs. (Prebleach: 3 frames at 2 s intevals; photobleach 3.74 s, postbleach acquisition, 20 frames at 2 s intervals, 20 frames at 5 s intervals and 10 frames at 20 s intervals).

Average fluorescence in the ROI was measured and background was subtracted. Decrease in fluorescence monitored in nearby reference ROIs was minimal under these conditions. Fluorescence intensity was normalized to baseline (average of all pre-bleach measurements) and plotted as a function of time.

We used two different methods to quantify the extent, and another two methods to quantify the kinetics, of post-bleach recovery of GFP-actin fluorescence. For all methods we first subtracted any remaining signal at the first post-bleach time point, and renormalized the data after setting this point as 0, as described previously (*Koskinen et al., 2014*).

For the first method to calculate the stable fraction of GFP-actin, we calculated the mean fluorescence of each trace during the period 250–300 s post-bleach, when recovery of the dynamic component fluorescence is essentially complete (*Koskinen et al., 2014*). This value was defined as the mobile fraction, and the stable fraction was defined as (1 − [mobile fraction]). All means per condition were then used for statistical comparison.

We also used a previously described method (*Honkura et al., 2008*; *Koskinen et al., 2014*) to calculate the stable fraction of GFP-actin. Briefly, after plotting the normalized values of each individual set of FRAP data, a linear extrapolation of each curve in the region exceeding approximately $5 \times t_{1/2 \text{ (dynamic)}}$ was made. The y intercept value was taken as the size of the mobile fraction, and the stable fraction was then calculated as above. Both these methods to determine the stable fraction of GFP-actin gave similar results.

To determine the recovery half-time, we used the same plots and determined the time point at which the dynamic component reached half the value of its maximal recovery (*Koskinen et al., 2014*). We also calculated recovery half-times by fitting data from the recovery phases of each individual trace to the equation $y(t) = y_0 + (\text{Plateau} - y_0) \times (1 - e^{-t/t_1})$ where $y_0$ is the fluorescence value immediately post-bleach, Plateau is the y value at infinite times and $t_1$ is the time constant of recovery (similar to the method of [*Koskinen and Hotulainen, 2014*]). Again, both methods to calculate the kinetics of recovery gave very similar results.

## FRAP of LIMK1-GFP

FRAP experiments for LIMK1-GFP were performed essentially as for GFP-actin, except that neurons were transfected with mCherry plus either wild type LIMK1-GFP or CCSS-LIMK1-GFP cDNAs. Traces were analyzed as for GFP-actin, except that half-times of recovery were only determined by the curve-fitting method, because the fluorescence signal for CCSS-LIMK1-GFP had not reached a plateau at the last time point examined. For the same reason, the linear extrapolation method was not used to determine the stable fraction for these experiments.

## Organotypic slice culture and biolistic transfection

Hippocampal sections (400 μm) were prepared from P7-8 Sprague Dawley rats using a MX-TS tissue slicer (Siskyou, Grants Pass, OR) and cultured on 0.4 μm cell culture inserts at 34°C as described (*Soares et al., 2013*). At DIV 8–10, neurons were transfected biolistically using a hand-held gene gun (Bio-Rad, Irvine, CA). For preparation of gene gun cartridges, 30–60 μg of DNA was added to 8–10 mg of 1.0 μm gold microcarriers (Biorad). Neurons were returned to the incubator for 3–4 days prior to imaging.

## Two photon glutamate uncaging

Transfected organotypic slices were transferred to an imaging chamber on a BX61WI upright microscope (60×/1.0 NA objective; Olympus) and continuously perfused at room temperature in ringer solution containing (in mM): 119 NaCl, 2.5 KCl, 0.1 MgSO$_4$, 3.0 CaCl$_2$, 1.0 NaH$_2$PO$_4$, 11 glucose, and 26.2 NaHCO$_3$, 0.01 glycine, 0.001 TTX and 2.5 MNI-Glutamate (Femtonics, Manassas, VA). For the morphological plasticity experiments, mCherry signal was imaged at 950 nm using a Ti: Sapphire pulsed laser (MaiTai-DeepSee, Spectra Physics, SantaClara, CA) and MNI-Glutamate was uncaged with a second laser tuned at 720 nm. Short segments of secondary and/or tertiary apical dendrites were continuously imaged at 1–2 min intervals by gathering z-stacks (0.75 μm steps) centered on the spine of interest. The glutamate uncaging protocol consisted of 4 ms pulses delivered at 0.5 Hz for 1 min (*Harvey and Svoboda, 2007*). Laser power of the uncaging beam was fixed across experiments at 30 mW, measured at the back aperture of the objective. Parallel electrophysiological

experiments (n > 50) showed that uncaging at this laser power consistently yields AMPAR-mediated inward currents between 5–30 pA. Experiments were performed on spines at approximately constant depth to minimize uneven light scattering between experiments.

To monitor changes in uncaging-evoked excitatory postsynaptic currents (uEPSCs) during single-spine LTP, CA1 neurons were voltage clamped at −70 mV with an internal solution containing (in mM): 115 cesium methane-sulfonate, 5 tetraethylammonium-Cl, 10 sodium phosphocreatine, 20 HEPES, 2.8 NaCl, 5 QX-314, 0.4 EGTA, 3 ATP($Mg^{2+}$ salt), and 0.5 GTP, and 0.02 Alexa 594 (pH 7.25, 280–290 mOsmol/l). Neurons were continuously perfused in a ringer solution containing (in mM): 119 NaCl, 2.5 KCl, 1.3 $MgSO_4$, 2.5 $CaCl_2$, 1.0 $NaH_2PO_4$, 11 glucose, and 26.2 $NaHCO_3$, 0.01 glycine, 0.001 TTX, and 2.5 mM MNI-Glutamate. Ti-sapphire lasers were tuned to 810 nm and 720 nm for imaging of Alexa 594 and uncaging of MNI-glutamate, respectively (*Soares et al., 2013*). Shortly after gaining whole cell access (<5 min), short duration light pulses (4 ms) were delivered at low frequency (0.05 Hz) to the tips of two adjacent dendritic spines to uncage MNI-Glutamate and establish a baseline uEPSC amplitude for each spine. Neurons were then voltage clamped at 0 mV and 30 consecutive light pulses were delivered to a single spine at 0.5 Hz to induce LTP (*Harvey and Svoboda, 2007*). This pairing protocol was performed less than 6 min after gaining whole cell access to avoid potential issues associated with washout (*Malinow and Tsien, 1990*). uEPSCs (at −70 mV) were then monitored for an additional 20 min at 0.05 Hz. Two-photon image stacks (810 nm) of the dendritic segment was sampled approximately every 5 min. To monitor changes in spine volume during these experiments, we measured the intensity of Alexa594 signal in both the stimulated and unstimulated spine. Because the intracellular dye concentration was typically not at a steady state under these conditions, we normalized spine intensities to the intensity of a nearby dendritic segment at each time point.

## Analysis of 2P uncaging-induced changes in spine volume

For MNI-glutamate uncaging experiments, a stackreg function in ImageJ (NIH) was used to align maximum intensity projected images from each time series to correct for X–Y drift. Changes in spine volume were estimated using an intensity-based method by summing pixel intensities from spine regions of interest at each time point, as described previously (*Harvey and Svoboda, 2007*). The average summed intensity of the first four baseline images (prior to uncaging) was used as $F_o$ and all intensity values are plotted as $\Delta F/F_o \times 100$. All summed intensity measurements were background subtracted and gathered from raw unprocessed images. Spine 'growth success' and spine 'shrinkage' events reflect instances where the average normalized intensity values remained above, or below, 2× the SD of the baseline (±16% for growth success and spine shrinkage, respectively) for the duration of the experiment after glutamate uncaging. Spine diameter measures (*Figure 5*) were estimated based on a full width at half maximum value (in µm), calculated by applying a Gaussian fit to the intensity profile across the spine head of the first baseline image.

## Quantification of synaptic puncta

Synaptic puncta in transfected neurons were defined as signals that were positive for GFP (morphology marker), PSD-95 (postsynaptic marker) and Synapsin (presynaptic marker). Synaptic puncta per unit length of dendrite were quantified using NIH ImageJ software. Briefly, PSD-95 and Synapsin images in each channel were thresholded by gray value at a level close to 50% of the dynamic range. This threshold value was kept constant for all images in each condition, and background noise from these images were negligible. The GFP fill was used to trace 100 µm segments along the three most prominent dendrites emanating from the cell body. Synaptic puncta of 2–20 pixel units, defined as above, were measured using the Boolean function 'AND' for the selected channels within each dendritic segment. Results were logged to a spreadsheet.

## Acknowledgements

We thank Dr C Benedict and S Karnam for molecular biological assistance, all Thomas lab members for helpful discussions and Drs G Gallo, Y Son (both SHPRC/Temple) and C Su (UCSD) for invaluable suggestions. Supported by seed funding from Shriners Hospitals for Children and NINDS grant R21NS087414 (both to GT). JG acknowledges a Postdoctoral Fellowship from Shriners Hospitals for Children.

# Additional information

## Funding

| Funder | Grant reference | Author |
| --- | --- | --- |
| National Institute of Neurological Disorders and Stroke (NINDS) | R21NS087414 | Gareth M Thomas |
| Shriners Hospitals for Children | | Gareth M Thomas |
| Shriners Hospitals for Children | Postdoctoral Fellowship | Joju George |

The funders had no role in study design, data collection and interpretation, or the decision to submit the work for publication.

## Author contributions

JG, CS, AM, Acquisition of data, Analysis and interpretation of data, Drafting or revising the article; J-CB, Analysis and interpretation of data, Drafting or revising the article; GMT, Conception and design, Acquisition of data, Analysis and interpretation of data, Drafting or revising the article

## Ethics

Animal experimentation: This study was performed in strict accordance with the recommendations in the Guide for the Care and Use of Laboratory Animals of the National Institutes of Health. All of the animals were handled according to approved institutional animal care and use committee (IACUC) protocols (#3439, #4277) of Temple University School of Medicine.

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
