## [Decision Letter]

Thank you for sending your work entitled “Palmitoylation of LIM Kinase-1 Ensures Spine-specific Actin Polymerization and Morphological Plasticity” for consideration at *eLife*. Your article has been favorably evaluated by a Senior editor and three reviewers, one of whom, Pekka Lappalainen, is a member of our Board of Reviewing Editors.

The Reviewing editor and the other reviewers discussed their comments before reaching the decision. As you will see, they all considered your findings on LIMK1 palmitoylation important and interesting, and found majority of the data convincing and of high technical quality. However, they stated that a few additional experiments are required to strengthen the conclusions presented, and that the manuscript text and some figures should be revised. Most importantly, it appears that palmitoylation has a more complex role in LIMK1 localization and function in spines than proposed in the present version of the manuscript. Furthermore, the model concerning cofilin activity gradient generated solely by LIMK1 activity was not found particularly convincing.

Major comments (required):

1) Does dipalmitoylation of LIMK1 simply function in membrane anchoring or does the membrane insertion of this region alter the interactions of the LIMK1 N-terminus with other binding partners? This could be tested by using a N-myr mutant protein with a single cysteine in the palmitoylation region (CS or SC) such that a doubly acylated N-terminal tail (myristate and palmitate) would be expressed in cells. Would this behave identically to the dipalmitoylated species? Unless the N-terminal myristoylation interferes with recognition of the palmitoylation domain by the PAT, this experiment should be easily done. If the Myr/Palm form accumulates to spines, is it activated normally by signaling through Pak?

2) The FRAP data presented in Figure 1—figure supplement 1 and Figure 5 should be properly fitted and analyzed. In addition to examining the sizes of immobile fractions, the authors should obtain information about the dynamics of the mobile fractions. Furthermore, it would be important to perform FRAP analysis also on GFP alone as a control for Figure 1—figure supplement 1, (i.e. to measure the dynamics of a truly soluble protein in spines). Overall, the data presented in Figure 1—figure supplement 1 suggest that both wild-type and CCSS-LIMK1 display very slow dynamics in spines, indicating that LIMK1 associates with other proteins that severely limit its diffusion at these sites. Thus, palmitoylation seems to play a more complex role in LIMK1 localization and function that just anchoring this protein to the plasma membrane. Moreover, the results of FRAP analysis presented in Figure 5 should be more precisely discussed by considering the known functions of LIMK1 and ADF/cofilins.

3) Actin dynamics and structural plasticity of dendritic spines have been associated with the maintenance and plasticity of “functional” strength of excitatory synapses. Demonstrating the importance of LIMK1 palmitoylation in the maintenance of excitatory synaptic transmission and glutamate uncaging-induced functional plasticity would thus significantly strengthen the manuscript.

Other comments:

1) The “shell-to-core” cofilin activity gradient hypothesis is not sufficiently well supported by the data in the present study. Therefore, the manuscript text and Figure 10 should be accordingly modified.

2) To provide the readers more balanced view on the mechanisms regulating cofilin activity in cells, it would be important to discuss the possible roles of phosphatases and other interacting proteins in regulating cofilin activity and actin dynamics in dendritic spines.

3) The palmitoyl acyltransferase data (Figure 4) are not particularly convincing, and should be either significantly strengthened or omitted from the manuscript.

4) The authors should discuss what lies upstream of LIMK1 palmitoylation in neurons.

Below, please also find the specific comments by the three reviewers. These will provide you additional information about how to deal with the points listed above.

*Reviewer #1*:

Precisely controlled actin filament assembly and disassembly are crucial for morphogenesis of dendritic spines in neurons. However, the spatial mechanisms controlling actin dynamics in spines are incompletely understood. Here, George et al. identify palmitoylation of LIMK1 as an important mechanism controlling actin dynamics in spines. They show that palmitoylation is critical for enrichment of LIMK1 (but not LIMK2) in spines and for its activation at these sites. Consequently LIMK1 palmitoylation is important for proper regulation of actin dynamics and long-term stability of dendritic spines.

The majority of the experiments presented appear to be of good technical quality, and the manuscript provides important new information concerning the spatial control of actin dynamics in neurons as well as on the mechanisms that control the subcellular localization of LIMK1. However, there are few important points that should be addressed to improve the study.

1) The results of FRAP experiments presented in Figure 5 are somewhat confusing, and the authors should perform better analysis of the FRAP data. From the data presented in Figure 5, it seems that that the actual rates of fluorescence recovery are quite similar in all cases (although this should be carefully examined by fitting the data). However, LIMK1 knockdown and CCSS-LIMK1 rescue cells exhibit a significantly larger immobile fraction of actin compared to control cells. Thus, while majority of actin filaments in control cells undergo relatively rapid turnover, approximately 30 % of actin filaments in LIMK1 (and CCSS-LIMK1 rescue spines) appear to be very stable. How does inactivation of LIMK1, which is a negative regulator of ADF/cofilin-mediated rapid actin filament disassembly, result in appearance of a stable actin filament pool in spines (because increased ADF/cofilin activity would be expected to increase (not decrease) actin filament turnover in spines)? This surprising result should be discussed in the 'Results' and 'Discussion' sections in more detail. Furthermore, the FRAP data should be properly fitted and analyzed as described e.g. in Koskinen et al., (Mol. Cell. Neurosci., 2014) to obtain information about the dynamics of the mobile actin fraction.

2) Similarly, the FRAP experiments shown in Figure 1—figure supplement 1 provided somewhat surprising results, and should be analyzed and discussed more carefully. Based on these experiments, it seems that both wild-type LIMK1 and CCSS-LIMK1 display very slow dynamics in spines. This indicates that LIMK1 associates in spines with other proteins that severely limit its mobility. To understand the dynamics of wild-type LIMK1 and CCSS-LIMK1 in spines, the authors should perform similar data fitting as requested for GFP-actin above as well as perform FRAP analysis on GFP alone as a control (i.e. to measure the dynamics of a truly soluble protein in spines). Overall, these data seem to suggest that palmitoylation is not required to immobilize LIMK1 in spines, but rather plays a more complex role in localizing this protein to dendritic spines and regulating its activity at these sites.

3) The data presented in Figure 4 are not particularly convincing and could perhaps be deleted from the manuscript. Is it possible that the differences in LIMK1 and NPY co-localization as well as in LIMK1 targeting to spines after brefeldin A treatment in proximal vs. distal regions could result from differences in the rate of depalmitoylation at these regions (instead of different palmitoyl acyltransferases interacting with LINK1 at different regions of dendrites)?

*Reviewer #2*:

This manuscript by George et al reports the role of LIMK1 palmitoylation in the regulation of actin polymerization and morphological plasticity in dendritic spines. In support of this, the authors show that LIMK1 is palmitoylated at the N-terminal double cysteines, and that this modification promotes spine localization of the protein. Functionally, this palmitoylation is required for actin polymerization and stability of dendritic spines, and also for activity-dependent spine volume increase. Lastly, LIMK1 palmitoylation promotes LIMK1 activation (T508 phosphorylation), which requires CaMKII and PAK activation.

This is an interesting identification of a novel synaptic protein that is modified by palmitoylation, and demonstration of a novel role of a protein palmitoylation in the regulation of synaptic actin polymerization and activity-dependent structural plasticity of dendritic spines. Given the importance of actin-regulatory mechanisms in dendritic spines, and the known involvement of LIMK1 in cognitive impairments in humans, this study seems to have a significant impact in the field.

1) This study mainly monitors actin polymerization and activity-dependent changes in spine volume as functional measures of the proposed mechanisms. However, actin dynamics and structural plasticity of dendritic spines have been keenly associated with the maintenance and plasticity of “functional” strength of excitatory synapses. Demonstrating the importance of LIMK1 palmitoylation in the maintenance of excitatory synaptic transmission and glutamate uncaging-induced functional plasticity would much strengthen the manuscript.

2) It is unclear what lies in the upstream of LIMK1 palmitoylation. Is NMDAR activation required? Is the activation of CaMKII or PAK required for LIMK1 palmitoylation? Which PATs are involved in LIMK1 palmitoylation? Is the T508 phosphorylation of LIMK1 required for its palmitoylation? These aspects are all unclear and should be explored to some extent or, less ideally, discussed. Some questions marks can be added to the summary diagram in Figure 10.

*Reviewer #3*:

Summary: In this very well written manuscript, the authors clearly demonstrate that the double cysteine motif at residues 7 and 8 of LIM kinase 1, which is not present in LIMK2, serves as a site for dipalmitoylation. This motif is both necessary and sufficient to target and anchor LIMK1 within dendritic spines. Interestingly, the authors find a brefeldin A-dependent and independent mechanism for spine delivery of palmitoylated-LIMK1, the former being used to supply proximal spines within about 100 μm of the soma and the latter used to deliver palmitoylated LIMK1 to distal spines. The activation of LIMK1 in spines by Pak is shown to be dependent upon LIMK1 palmitoylation, although in vitro, soluble active Pak can phosphorylate wild type and the non-palmitoylatable mutant CCSS-LIMK1 equally well. Studies on actin assembly dynamics using fluorescence recovery after photobleaching (FRAP) in spines of neurons expressing WT and mutant forms of LIMK1 show that normal dynamics and spine enlargement driven by glutamate stimulation requires palmitoylated LIMK1.

Critique: The sound experiments presented and the high quality results shown in the figures require very little comment. However there are some results that are missing which would be welcome additions to what is presented.

A question remains as to the role of the dipalmitoylation—is it simply membrane anchoring or does the membrane insertion of this region alter the interactions with other binding partners of the LIMK1 N-terminus? I am surprised that the authors did not use the N-myr mutant protein with a single cysteine in the palmitoylation region (CS or SC) such that a doubly acylated N-terminal tail (myristate and palmitate) resulted. Would this behave identically to the dipalmitoylated species? Unless the N-terminal myristoylation interfered with recognition of the palmitoylation domain by the PAT, this experiment should be easily done. If the Myr/Palm form is bound to membrane in spines, is it activated normally by signaling through Pak?

The evidence for the Golgi-dependence for proximal spine delivery of the palmitoylated LIMK1 is very strong, but the evidence for distal spine delivery requiring localized PAT activity is speculative. Although this mechanism is certainly possible, it may not be the only possibility for distal delivery. Until somatodendritic PATs can be down regulated differentially from those at the Golgi, it may remain a speculation.

In spite of the very high quality work shown in the studies presented here, there are still many unanswered questions regarding the model put forward to explain the centripetal retrograde flow of actin subunits that are adding onto actin filaments at the spine periphery and disassembling in the center. In this regard it is incorrect to talk about cofilin as “severing filament pointed ends” (in the subsection headed “Polarized “shell-to-core” actin regulation in spines by palmitoyl-LIMK1”) and referencing the 2003 Pollard and Borisy review. Severing can occur anywhere in the filament where ADP-actin subunits predominate (which is generally more toward the pointed end of dynamic filaments). However, the type of turnover of these branched networks that is occurring is more correctly termed array treadmilling and there are many more current reviews, one of which should be cited for this. Given that the authors favor a cofilin gradient model, it is surprising that the Discussion fails to include a reference to work published from the Zheng lab in Nature Neuroscience (Gu J, et al., Nat Neurosci 13, 1208-1215, 2010) in which cofilin inhibition by LIMK1 was required for LTP-induced spine enlargement, a role proposed here for the LIMK1 in spines.

The speculative model that focuses on the possibility of a cofilin activity gradient also requires some comment. There are many additional factors that need to be considered. First, for cofilin to be regulated in such a manner its concentration would have to be quite low in spines and one would expect the phosphatases, which regulate its activation, would also need to be localized to the central domain because their activity seems to dominate the phosphorylation state of cofilin (see Gu paper referenced above and others). There is evidence that slingshot-1L, a major cofilin phosphatase is only active when bound to F-actin, a fact that could help support the gradient model (Nagata-Ohashi K, et al. J Cell Biol. 165: 465-471, 2004: Soosairajah J, et al., EMBO J 24, 473-486, 2005). However, other studies suggest that the actin filament severing activity of cofilin is modulated (enhanced) in vivo by Aip1 (see recent Chen et al., J Biol Chem, 290: 2289-2300, 2015 and references theirin) and that cofilin's recruitment to F-actin is also modulated by coronin 1A (Kueh HY et al., J Cell Biol. 182:341-353, 2008). These proteins likely impact the actin dynamics and retrograde flow. Finally, many (perhaps all?) spines may undergo transient penetration by microtubules (Merriam EB, et al., J Neurosci 33: 16471-82, 2013)), whose assembly may also be modulated in a LIMK1-dependent manner by p25/TPPP, a protein that affects HDAC activity and acetylation of MTs (Acevedo K, et al., Exp Cell Res, 313, 4091-4106, 2007; Schofield AV, et al., J Biol Chem 288: 7907-17, 2013). I am not trying to argue for any one of these models, but simply believe that in this paper the authors have not convinced me that a gradient of cofilin activity exists nor that cofilin is the only target involved in regulating the actin dynamics. Maybe tone this model down a bit and remove the figure since such a figure is often what the reader will take away, even if totally speculative.

---

## [Author Response]

*1) Does dipalmitoylation of LIMK1 simply function in membrane anchoring or does the membrane insertion of this region alter the interactions of the LIMK1 N-terminus with other binding partners? This could be tested by using a N-myr mutant protein with a single cysteine in the palmitoylation region (CS or SC) such that a doubly acylated N-terminal tail (myristate and palmitate) would be expressed in cells. Would this behave identically to the dipalmitoylated species? Unless the N-terminal myristoylation interferes with recognition of the palmitoylation domain by the PAT, this experiment should be easily done. If the Myr/Palm form accumulates to spines, is it activated normally by signaling through Pak*?

We agree with the reviewers that analyzing LIMK1 mutants modified with myristate plus palmitate is an excellent way to gain more insight into the role of LIMK1 palmitoylation. As suggested, we therefore generated Myr-C7S and Myr-C8S LIMK1 mutants and assessed their localization and activation status in hippocampal neurons. Interestingly, neither mutant recapitulated the spine enrichment of wild type LIMK1, and neither mutant was as efficiently phosphorylated at T508. We have included these additional localization experiments in a new Figure 2—figure supplement 2 and have added the phosphorylation data to a modified Figure 7 (renumbered following removal of Figure 4, as per the reviewers’ suggestion). We mention in the legend to Figure 7 that phosphorylation of Myr-C8S-LIMK1 differs significantly from that of Myr-CCSS-LIMK1, suggesting that, at least for this mutant, the myristoylation sequence does not interfere with recognition by the PAT. Taken together, these results suggest that myristate + palmitate modification is not equivalent to dipalmitoylation, and that an intact CC di-palmitoylation motif is essential to target LIMK1 to spines. Based on our accompanying pharmacological data (Figure 8), the spine enrichment of key ‘upstream’ kinases (activated forms of PAK and CaMKII) likely explains why LIMK1 mutants in other locations are not activated. We have modified the Discussion to include these new findings and to summarize how dipalmitoylation controls LIMK1 localization and signaling in neurons.

*2) The FRAP data presented in*
Figure 1—figure supplement 1
*and*
Figure 5
*should be properly fitted and analyzed. In addition to examining the sizes of immobile fractions, the authors should obtain information about the dynamics of the mobile fractions. Furthermore, it would be important to perform FRAP analysis also on GFP alone as a control for*
Figure 1—figure supplement 1*, (i.e. to measure the dynamics of a truly soluble protein in spines). Overall, the data presented in*
Figure 1—figure supplement 1
*suggest that both wild-type and CCSS-LIMK1 display very slow dynamics in spines, indicating that LIMK1 associates with other proteins that severely limit its diffusion at these sites. Thus, palmitoylation seems to play a more complex role in LIMK1 localization and function that just anchoring this protein to the plasma membrane. Moreover, the results of FRAP analysis presented in*
Figure 5
*should be more precisely discussed by considering the known functions of LIMK1 and ADF/cofilins*.

We appreciate the need to more fully analyze our FRAP data and now include these additional analyses in updated versions of Figure 1—figure supplement 1 (for LIMK1-GFP) and in both Figure 4, and a new Figure 4—figure supplement 2 (for GFP-actin; again, please note the adjusted numbering following removal of our original Figure 4).

For the GFP-actin experiments, we performed two analyses to determine the size of the stable fraction, both of which reinforced our initial conclusion that loss of palmitoyl-LIMK1 increases the fraction of stable actin. We also used two analytical methods to determine the kinetics of recovery of the dynamic fraction, both of which revealed that this is not significantly affected under any condition examined. In addition to updating the Figures as described above, we have modified the Methods section to include details of these additional analyses, which benefitted greatly from the reference kindly provided by Reviewer #1 (35).

Regarding our LIMK1-GFP FRAP experiments, we agree with the reviewer that both wild type and CCSS-LIMK1-GFP display very slow dynamics in spines. Our new analysis of the stable fraction of LIMK1-GFP in spines (modified Figure 1—figure supplement 1) supports the conclusion that CCSS mutation decreases the fraction of stable LIMK1. We also quantified the recovery times of the mobile pool for each LIMK1-GFP construct. Although the lack of wild type LIMK1-GFP fluorescence recovery makes determining the precise recovery kinetics difficult, (a point explained in the legend to Figure 1—figure supplement 1) both wt- and CCSS-LIMK1-GFP clearly recover markedly more slowly than freely soluble cytosolic proteins. We thus fully agree with the reviewer that these results suggest that LIMK1 associates with other proteins that limit its diffusion. Thus, although the effect of palmitoylation on stabilizing LIMK1 in spines is significant, it may be less critical than its roles in spine targeting (Figure 1), and in ensuring LIMK1 phosphorylation by spine-specific upstream activators (Figures 7 and 8). We have modified our Discussion to include these points.

We also attempted to measure the kinetics of recovery of cytosolic GFP alone, but using the parameters for our GFP-actin and LIMK1-GFP FRAP experiments we found that cytosolic GFP fluorescence recovered too rapidly for us to accurately plot recovery curves. In the limited time available for revision we were unfortunately unable to re-optimize conditions for this experiment, but we note that several reports (e.g. [62], Nat Neurosci; Sharma et al., 2006, Mol. Cell. Neurosci; [76], J. Neurosci) have already demonstrated very rapid FRAP for cytosolic GFP in dendritic spines (time constant <1 sec; [62]). Moreover, another report ([8], Cell 140, 567-578) found that FRAP of GFP-tagged Rpt1, a protein of similar size (75 kDa) to LIMK1-GFP (95 kDa) was also extremely rapid. We have modified both our Results and Discussion to include these points and note that they all support the reviewer’s hypothesis that LIMK1 does not behave like a soluble protein and that other factors, most likely protein-protein interactions, markedly limit LIMK1 diffusion in spines.

Finally, in response to both this comment and to further points raised by Reviewers #1 and #3, we have considered our GFP-actin FRAP results with regard to known roles of LIMK1 and ADF/cofilins. We agree that the increase in immobile actin in the absence of palmitoyl-LIMK1 is surprising, given LIMK1’s known roles. We have therefore modified our Results to highlight this finding and have included three possible explanations in the Discussion, which we summarize here:

One explanation for our results could arise from the ability of active (dephospho-) cofilin to induce a twist in actin filament conformation that propagates along the filament (e.g. [5], Trends Cell Biol.). Filaments are more likely to sever at boundaries between cofilin-bound (twisted) and unbound (untwisted) regions ([9], J. Mol. Biol.). However, an entire cofilin-bound filament, as might occur in the absence of LIMK1, is more stable than a bare filament ([18], Biophys Chem; [2], Mol. Cell). Thus, if cofilin phosphorylation is reduced in LIMK1 knockdown (and CCSS-LIMK1 ‘rescue’) spines, then these spines may contain a subset of filaments that are decorated with active cofilin but which are actually more stable.

A second, related possibility is that increased levels of dephospho-cofilin, caused by loss of palmitoyl-LIMK1, may enhance formation of cofilin-actin rods. Rods are stable structures that are only formed by dephospho-cofilin and actin turnover within rods is very slow ([7], Am. J. Physiol. Cell Physiol.), which could account for the increased stable GFP-actin that we observe. However, we note that cofilin-actin rods have been documented in neurites but not, to our knowledge, within spines.

A third, intriguing explanation, which we favor, arises from a prior report that LTD-like stimulation (low frequency electrical stimulation or bath application of NMDA) markedly increases actin filament stability in spines ([62], Nat Neurosci). Strikingly, as noted in our initial manuscript, the LTP-like stimuli used in our uncaging experiments result in shrinkage of a significant subset of spines (an LTD-like effect) in the LIMK1 knockdown and CCSS-LIMK1 rescue conditions (Figure 5—figure supplement 1). This suggests that ‘learning rules’ are impaired in some LIMK1 knockdown (and CCSS-LIMK1 rescue) spines, so that they respond to LTP-like inputs with LTD-like outputs. Spontaneous bursts of activity in our cultured neurons may therefore be interpreted as LTD-like stimuli by some LIMK1 knockdown spines. In the short term this would be predicted to decrease actin filament turnover, as reported by Star et al., while over longer times this impaired actin turnover may contribute to the spine shrinkage and synapse loss that we observe. We note in our revised Discussion that the three explanations above are not mutually exclusive and could all contribute to the impaired actin filament turnover in the absence of palmitoyl-LIMK1. Moreover, we also mention the possibility that other potential LIMK1 substrates, in addition to cofilin, may contribute to this effect.

*3) Actin dynamics and structural plasticity of dendritic spines have been associated with the maintenance and plasticity of* “*functional*” *strength of excitatory synapses. Demonstrating the importance of LIMK1 palmitoylation in the maintenance of excitatory synaptic transmission and glutamate uncaging-induced functional plasticity would thus significantly strengthen the manuscript*.

We appreciate and share the reviewers’ interest in palmitoyl-LIMK1’s role in the regulation of synaptic transmission and functional plasticity. We therefore performed new experiments to address whether the activity-dependent structural plasticity, which is palmitoyl-LIMK1-dependent (Figure 5), is accompanied by changes in functional plasticity. In a new Figure 5—figure supplement 2 we now include important proof-of-principle experiments that confirm that our 2-photon uncaging stimulation reliably increases not only spine volume but also AMPA uncaging-evoked postsynaptic currents (uEPSCs), and that both changes are specific to the stimulated spine. Importantly, previous studies reported very similar molecular requirements for uncaging-induced structural and functional plasticity, and in particular identified key roles for both actin polymerization and Cdc42 signaling in both processes ([42], Nature; [46], Nature). Our new experiment therefore supports the hypothesis that LIMK1 palmitoylation may also be required for activity-dependent functional plasticity, and may be a key link between the Cdc42 signaling and actin polymerization described by others. We have modified our Discussion to include these important points. Finally, we note that in several studies that focused on mechanisms of spine cytoskeletal regulation using 2-photon uncaging (e.g. [11], Neuron; Steiner et al., 2008, Neuron; Tanaka et al., 2008, Science), functional plasticity experiments were limited to confirming that the same uncaging protocol induces both structural and functional plasticity, as we now demonstrate.

*Other comments*:

*1) The* “*shell-to-core*” *cofilin activity gradient hypothesis is not sufficiently well supported by the data in the present study. Therefore, the manuscript text and Figure 10 should be accordingly modified*.

We appreciate this concern of the reviewers and have therefore modified the text of the Discussion accordingly. We also discuss other factors that may influence spatial control of filament polymerization/ depolymerization by palmitoyl-LIMK1, including other proteins that bind and/or affect activity of cofilin and/or additional LIMK1 substrates. Our discussion of these issues benefits greatly from (but is not limited to) the references kindly provided by Reviewer #3, and we expand on these points in our reply to this reviewer below. In addition, we have modified Figure 10 to remove the gradient of cofilin activity, though we retained the schematic showing membrane-specific activation of palmitoyl-LIMK1.

*2) To provide the readers more balanced view on the mechanisms regulating cofilin activity in cells, it would be important to discuss the possible roles of phosphatases and other interacting proteins in regulating cofilin activity and actin dynamics in dendritic spines*.

As described in the reply to Point #1 above, we agree with the need to discuss other regulators of cofilin activity and interactions and have modified the Discussion to include these points.

*3) The palmitoyl acyltransferase data (*Figure 4*) are not particularly convincing, and should be either significantly strengthened or omitted from the manuscript*.

To address this comment and to gain more insight into how LIMK1 palmitoylation is controlled, we performed a preliminary screen of all 23 mouse PATs (see also reply to Reviewer #2 below). We identified both Golgi-localized PATs and synaptodendritic PATs that can palmitoylate LIMK1, a finding that is broadly consistent with our original hypothesis that LIMK1 palmitoylation occurs in both of these locations. However, we prefer not to include these data as we realize that significantly more work will be required to properly define how LIMK1 palmitoylation is controlled in neurons. We have therefore followed the reviewers’ suggestion to remove Figure 4, and will address this question fully in future studies.

*4) The authors should discuss what lies upstream of LIMK1 palmitoylation in neurons*.

We agree that this point is worthy of further discussion and have modified the Discussion text to address it. We both summarize our PAT screening findings (in reply to Point #3 above) and also discuss possible regulation of LIMK1 palmitoylation by neuronal activity or other factors. We mention additional preliminary findings that Bicuculline and KCl treatments do not increase LIMK1 palmitoylation, suggesting that LIMK1 is palmitoylation is likely constitutive. However, we also stress potential caveats regarding this conclusion, in particular the possibility that other stimuli could alter LIMK1 palmitoylation e.g. ephrins (which rapidly affect spine morphology via PAK) and/or that technical limitations of the ABE assay (which detects the entire cellular complement of palmitoyl-LIMK1, irrespective of location) could mask selective changes in LIMK1 palmitoylation at spines.

*Below, please also find the specific comments by the three reviewers. These will provide you additional information about how to deal with the points listed above*.

Reviewer #1:

*1) The results of FRAP experiments presented in*
Figure 5
*are somewhat confusing, and the authors should perform better analysis of the FRAP data. From the data presented in*
Figure 5*, it seems that that the actual rates of fluorescence recovery are quite similar in all cases (although this should be carefully examined by fitting the data). However, LIMK1 knockdown and CCSS-LIMK1 rescue cells exhibit a significantly larger immobile fraction of actin compared to control cells. Thus, while majority of actin filaments in control cells undergo relatively rapid turnover, approximately 30 % of actin filaments in LIMK1 (and CCSS-LIMK1 rescue spines) appear to be very stable. How does inactivation of LIMK1, which is a negative regulator of ADF/cofilin-mediated rapid actin filament disassembly, result in appearance of a stable actin filament pool in spines (because increased ADF/cofilin activity would be expected to increase (not decrease) actin filament turnover in spines)? This surprising result should be discussed in the 'Results' and 'Discussion' sections in more detail. Furthermore, the FRAP data should be properly fitted and analyzed as described e.g. in Koskinen et al., (Mol. Cell. Neurosci., 2014) to obtain information about the dynamics of the mobile actin fraction*.

As described in our reply to the Major Comments, we appreciate the reviewer’s concern and have analyzed our GFP-actin FRAP data accordingly. The additional analyses confirm the reviewer’s assessment that, in both LIMK1 ‘knockdown’ and CCSS-LIMK1 ‘rescue’ neurons, there is an increased percentage of stable GFP-actin, but no significant effect on the kinetics of the remaining mobile GFP-actin. We have added these data in a new Figure 4—figure supplement 2 (renumbered following removal of our original Figure 4, as per reviewers’ suggestions) and include details of these analyses in the Methods.

We agree with the Reviewer that the increase in immobile actin is surprising, given the known roles of LIMK1. We have therefore modified both the Results and Discussion to highlight this finding and to include three possible explanations for this result, which we summarize here:

One possible explanation arises from the ability of active (dephospho-) cofilin to induce a twist in actin filament conformation that propagates along the filament (e.g. [5], Trends Cell Biol.). Filaments are more likely to sever at boundaries between cofilin-bound (twisted) and unbound (untwisted) regions ([9], J. Mol. Biol.). However, an entire cofilin-bound filament, as might occur in the absence of LIMK1, is more stable than a bare filament ([18], Biophys Chem; [2], Mol. Cell). Thus, LIMK1 knockdown (and CCSS-LIMK1 ‘rescue’) spines may contain a subset of filaments that are decorated with active cofilin but which are actually more stable.

A second possibility is that LIMK1 knockdown (which is predicted to lead to increased levels of dephospho-cofilin) enhances formation of cofilin-actin rod. Rods are stable structures that are only formed by dephospho-cofilin amd actin turnover within rods is very slow ([7], Am. J. Physiol. Cell Physiol.). However, although this explanation could account for the reduced mobility of GFP-actin that we observe, we note that cofilin-actin rods have been documented in neurites but not, to our knowledge, within spines.

A third, intriguing explanation, which we favor, arises from a prior report that LTD-like stimuli (low frequency electrical stimulation or bath application of NMDA) markedly increase actin filament stability in spines ([62], Nat Neurosci). Strikingly, as noted in our initial manuscript, the LTP-like stimulus used in our uncaging experiments usually triggers spine enlargement, but in a subset of LIMK1 knockdown and CCSS-LIMK1 rescue spines, instead triggers spine shrinkage (Figure 5—figure supplement 1). This suggests that some LIMK1 knockdown (and CCSS-LIMK1 rescue) spines respond to LTP-like inputs with LTD-like outputs i.e. that their learning rules are impaired. Spontaneous bursts of activity in our cultured neurons may therefore be interpreted as an LTD-like stimulus by some LIMK1 knockdown spines. In the short term this may result in decreased actin filament turnover, as reported by Star et al., while over longer times this impaired actin turnover may contribute to the spine shrinkage and synapse loss seen in the absence of palmitoyl-LIMK1. We note that the three explanations above are not mutually exclusive and could all contribute to the impaired actin filament turnover in the absence of palmitoyl-LIMK1.

*2) Similarly, the FRAP experiments shown in*
Figure 1—figure supplement 1
*provided somewhat surprising results, and should be analyzed and discussed more carefully. Based on these experiments, it seems that both wild-type LIMK1 and CCSS-LIMK1 display very slow dynamics in spines. This indicates that LIMK1 associates in spines with other proteins that severely limit its mobility. To understand the dynamics of wild-type LIMK1 and CCSS-LIMK1 in spines, the authors should perform similar data fitting as requested for GFP-actin above as well as perform FRAP analysis on GFP alone as a control (i.e. to measure the dynamics of a truly soluble protein in spines). Overall, these data seem to suggest that palmitoylation is not required to immobilize LIMK1 in spines, but rather plays a more complex role in localizing this protein to dendritic spines and regulating its activity at these sites*.

This is another insightful point by the Reviewer, which we have also addressed in our reply to the Major Comments. To summarize briefly, we have expanded Figure 1—figure supplement 1 to include quantitative measurements of the size of the stable fractions and the dynamics of the mobile fractions, for both wt- and CCSS-LIMK1-GFP. We agree with the reviewer that the slow dynamics of CCSS-LIMK1 recovery suggest that LIMK1 associates with other proteins that limit its diffusion and have modified our Discussion to include this point. In particular, based on the markedly different targeting and phosphorylation of wild type LIMK1, compared to myr-SC and myr-CS LIMK1, we emphasize that palmitoylation is critical for targeting LIMK1 to spines and does not act as a simple membrane anchoring signal.

As mentioned in our reply to the Major Comments, we found that soluble GFP fluorescence recovers too rapidly for us to plot recovery curves under the conditions used for our GFP-actin and LIMK1-GFP experiments. However, we note that several published studies (e.g. [62], Nat Neurosci; Sharma et al., 2006, Mol. Cell. Neurosci; [76], J. Neurosci) have already documented this extremely fast fluorescence recovery for soluble GFP in dendritic spines. We have included this point in our Results section and noted that it supports the reviewer’s conclusion that other factors, most likely protein-protein interactions, markedly limit LIMK1 diffusion in spines.

*3) The data presented in*
Figure 4
*are not particularly convincing and could perhaps be deleted from the manuscript. Is it possible that the differences in LIMK1 and NPY co-localization as well as in LIMK1 targeting to spines after brefeldin A treatment in proximal vs. distal regions could result from differences in the rate of depalmitoylation at these regions (instead of different palmitoyl acyltransferases interacting with LINK1 at different regions of dendrites)*?

We agree with the Reviewer that our conclusions based on the data in Figure 4 were somewhat premature. The possibility that differential thioesterase (as opposed to PAT) activity could explain these results is an insightful observation that we had not considered. As a result we have decided to follow the Reviewer’s suggestion to remove Figure 4 and will seek to better define the mechanisms that control LIMK1 palmitoylation in future work.

Reviewer #2:

*This manuscript by George et al reports the role of LIMK1 palmitoylation in the 1) This study mainly monitors actin polymerization and activity-dependent changes in spine volume as functional measures of the proposed mechanisms. However, actin dynamics and structural plasticity of dendritic spines have been keenly associated with the maintenance and plasticity of* “*functional*” *strength of excitatory synapses. Demonstrating the importance of LIMK1 palmitoylation in the maintenance of excitatory synaptic transmission and glutamate uncaging-induced functional plasticity would much strengthen the manuscript*.

We agree with the reviewer that the possibility that palmitoyl-LIMK1 also controls the functional strength and/or plasticity of excitatory synapses is an exciting one. We therefore performed new experiments to address whether the activity-dependent structural plasticity, which is palmitoyl-LIMK1-dependent (Figure 5), is accompanied by changes in functional plasticity. In a new Figure 5—figure supplement 2 we now include important proof-of-principle experiments that confirm that our 2-photon uncaging stimulation reliably increases not only spine volume but also AMPA uncaging-evoked postsynaptic currents (uEPSCs), and that both changes are specific to the stimulated spine. Importantly, previous studies reported very similar molecular requirements for spine-specific structural and functional plasticity, and in particular identified key roles for actin polymerization and Cdc42 signaling in both processes ([42], Nature; [46], Nature). Our new experiment therefore supports the hypothesis that LIMK1 palmitoylation may also be required for activity-dependent functional plasticity, and may be a key link between the Cdc42 signaling and actin polymerization described by others. We have modified our Discussion to include these important points.

We are interested to further pursue how palmitoyl-LIMK1 might regulate functional plasticity e.g. by affecting different populations of neurotransmitter receptors, but are aware that this question differs from the focus of our current manuscript on spatial control of actin dynamics. We therefore prefer to fully address this issue in future work, and note that in several studies that focused on mechanisms of spine cytoskeletal regulation using 2-photon uncaging (e.g. [11], Neuron; Steiner et al., 2008, Neuron; Tanaka et al., 2008, Science), functional plasticity experiments were limited to confirming that the same uncaging protocol induces both structural and functional plasticity, as we now demonstrate.

*2) It is unclear what lies in the upstream of LIMK1 palmitoylation. Is NMDAR activation required? Is the activation of CaMKII or PAK required for LIMK1 palmitoylation? Which PATs are involved in LIMK1 palmitoylation? Is the T508 phosphorylation of LIMK1 required for its palmitoylation? These aspects are all unclear and should be explored to some extent or, less ideally, discussed. Some questions marks can be added to the summary diagram in Figure 10*.

We appreciate the importance of all of these questions. We previously assessed neuronal LIMK1 palmitoylation levels by ABE assay, but observed no changes following treatment with either Bicuculline or KCl. These results, while preliminary, suggest that LIMK1 palmitoylation is largely constitutive and that other aspects of LIMK1 regulation e.g. its phosphorylation by PAK, which is likely CaMKII-dependent, are more likely to be the key activity-regulated steps in neurons. However, as we detect the entire cellular complement of palmitoyl-LIMK1 in ABE assays, irrespective of subcellular location, it is still possible that synaptic activity or other stimuli alter palmitoylation of a specific subcellular pool of LIMK1. We have added these points to the Discussion.

We also comprehensively assessed, as described in our response to the overall Other Comments, the PATs that can palmitoylate LIMK1 in cotransfected HEK293T cells, but found that several PATs can increase LIMK1 palmitoylation. More work will therefore be required to determine how LIMK1 palmitoylation is controlled in neurons. Again, we have briefly noted these preliminary findings in the Discussion, and are excited to fully address this issue in future studies.

Reviewer #3:

*Critique: The sound experiments presented and the high quality results shown in the figures require very little comment. However there are some results that are missing which would be welcome additions to what is presented*.

*A question remains as to the role of the dipalmitoylation— is it simply membrane anchoring or does the membrane insertion of this region alter the interactions with other binding partners of the LIMK1 N-terminus? I am surprised that the authors did not use the N-myr mutant protein with a single cysteine in the palmitoylation region (CS or SC) such that a doubly acylated N-terminal tail (myristate and palmitate) resulted. Would this behave identically to the dipalmitoylated species? Unless the N-terminal myristoylation interfered with recognition of the palmitoylation domain by the PAT, this experiment should be easily done. If the Myr/Palm form is bound to membrane in spines, is it activated normally by signaling through Pak*?

We appreciate the reviewer’s interest in the role of the dipalmitoylation of LIMK1 and agree that examining the targeting and regulation of a “myristoyl + palmitoyl” LIMK1 is an excellent way to gain more insight into this issue. We therefore generated both Myr-SC- and Myr-CS-LIMK1 mutants and examined their localization and activation status in neurons. Interestingly, neither mutant recapitulates the spine enrichment or activation status of wild type LIMK1. We have included the additional localization data in a new Figure 2—figure supplement 2, modified Figure 7 to include the additional phosphorylation data, and have added an additional statement regarding the unique role of dipalmitoylation to the Discussion. We also include a brief statement that the phosphorylation of Myr-CS-LIMK1 differs from either SC-LIMK1, CS-LIMK1 or myr-CCSS-LIMK1. This result suggests that, at least for Myr-CS-LIMK1, the addition of the myristolyation tag does not interfere with recognition of the remaining palmitoyl-site, but that myristate+palmitate is not equivalent to di-palmitoylation.

*The evidence for the Golgi-dependence for proximal spine delivery of the palmitoylated LIMK1 is very strong, but the evidence for distal spine delivery requiring localized PAT activity is speculative. Although this mechanism is certainly possible, it may not be the only possibility for distal delivery. Until somatodendritic PATs can be down regulated differentially from those at the Golgi, it may remain a speculation*.

We agree with the reviewer that our current evidence is insufficient to conclude that localized PAT activity controls distal spine delivery of palmitoyl-LIMK1. This point was also raised by Reviewer #1, who suggested that Figure 4 could therefore be removed. We have decided to follow this suggestion and to fully address the control of LIMK1 palmitoylation in future studies.

*In spite of the very high quality work shown in the studies presented here, there are still many unanswered questions regarding the model put forward to explain the centripetal retrograde flow of actin subunits that are adding onto actin filaments at the spine periphery and disassembling in the center. In this regard it is incorrect to talk about cofilin as* “*severing filament pointed ends*” *(in the subsection headed “Polarized “shell-to-core” actin regulation in spines by palmitoyl-LIMK1”) and referencing the 2003 Pollard and Borisy review. Severing can occur anywhere in the filament where ADP-actin subunits predominate (which is generally more toward the pointed end of dynamic filaments). However, the type of turnover of these branched networks that is occurring is more correctly termed array treadmilling and there are many more current reviews, one of which should be cited for this. Given that the authors favor a cofilin gradient model, it is surprising that the Discussion fails to include a reference to work published from the Zheng lab in Nature Neuroscience (Gu J, et al., Nat Neurosci 13, 1208-1215, 2010) in which cofilin inhibition by LIMK1 was required for LTP-induced spine enlargement, a role proposed here for the LIMK1 in spines*.

*The speculative model that focuses on the possibility of a cofilin activity gradient also requires some comment. There are many additional factors that need to be considered. First, for cofilin to be regulated in such a manner its concentration would have to be quite low in spines and one would expect the phosphatases, which regulate its activation, would also need to be localized to the central domain because their activity seems to dominate the phosphorylation state of cofilin (see Gu paper referenced above and others). There is evidence that slingshot-1L, a major cofilin phosphatase is only active when bound to F-actin, a fact that could help support the gradient model (Nagata-Ohashi K, et al. J Cell Biol. 165: 465-471, 2004: Soosairajah J, et al., EMBO J 24, 473-486, 2005). However, other studies suggest that the actin filament severing activity of cofilin is modulated (enhanced) in vivo by Aip1 (see recent Chen et al., J Biol Chem, 290: 2289-2300, 2015 and references theirin) and that cofilin's recruitment to F-actin is also modulated by coronin 1A (Kueh HY et al., J Cell Biol. 182:341-353, 2008). These proteins likely impact the actin dynamics and retrograde flow. Finally, many (perhaps all?) spines may undergo transient penetration by microtubules (Merriam EB, et al., J Neurosci 33: 16471-82, 2013)), whose assembly may also be modulated in a LIMK1-dependent manner by p25/TPPP, a protein that affects HDAC activity and acetylation of MTs (Acevedo K, et al., Exp Cell Res, 313, 4091-4106, 2007; Schofield AV, et al., J Biol Chem 288: 7907-17, 2013). I am not trying to argue for any one of these models, but simply believe that in this paper the authors have not convinced me that a gradient of cofilin activity exists nor that cofilin is the only target involved in regulating the actin dynamics. Maybe tone this model down a bit and remove the figure since such a figure is often what the reader will take away, even if totally speculative*.

These are all excellent points and we thank the reviewer for this very informative summary of possible alternative explanations and hypotheses. We have markedly modified our Discussion of sub-spine regulation by LIMK1 to reference the supportive studies mentioned by the reviewer, but also to include the alternative/additional explanations that s/he describes. We have taken care to note that our thoughts on intra-spine actin regulation by palmitoyl-LIMK1 are speculative and have modified Figure 10 to remove the cofilin gradient model as suggested.